# STAG2 promotes the myelination transcriptional program in oligodendrocytes

**Ningyan Cheng[1], Guanchen Li[2,3,4], Mohammed Kanchwala[5], Bret M Evers[6], Chao Xing[5,7], Hongtao Yu[2,3,4]\***

[1]Department of Pharmacology, University of Texas Southwestern Medical Center, Dallas, United States; [2]Westlake Laboratory of Life Sciences and Biomedicine, Hangzhou, China; [3]School of Life Sciences, Westlake University, Hangzhou, China; [4]Institute of Biology, Westlake Institute for Advanced Study, Hangzhou, China; [5]Eugene McDermott Center for Human Growth and Development, University of Texas Southwestern Medical Center, Dallas, United States; [6]Division of Neuropathology, University of Texas Southwestern Medical Center, Dallas, United States; [7]Department of Bioinformatics, Department of Population and Data Sciences, University of Texas Southwestern Medical Center, Dallas, United States

**\*For correspondence:**
yuhongtao@westlake.edu.cn

**Competing interest:** The authors declare that no competing interests exist.

**Abstract** Cohesin folds chromosomes via DNA loop extrusion. Cohesin-mediated chromosome loops regulate transcription by shaping long-range enhancer–promoter interactions, among other mechanisms. Mutations of cohesin subunits and regulators cause human developmental diseases termed cohesinopathy. Vertebrate cohesin consists of SMC1, SMC3, RAD21, and either STAG1 or STAG2. To probe the physiological functions of cohesin, we created conditional knockout (cKO) mice with *Stag2* deleted in the nervous system. *Stag2* cKO mice exhibit growth retardation, neurological defects, and premature death, in part due to insufficient myelination of nerve fibers. *Stag2* cKO oligodendrocytes exhibit delayed maturation and downregulation of myelination-related genes. *Stag2* loss reduces promoter-anchored loops at downregulated genes in oligodendrocytes. Thus, STAG2-cohesin generates promoter-anchored loops at myelination-promoting genes to facilitate their transcription. Our study implicates defective myelination as a contributing factor to cohesinopathy and establishes oligodendrocytes as a relevant cell type to explore the mechanisms by which cohesin regulates transcription.

## Editor's evaluation

This manuscript will be of interest to scientists working on genome organisation and transcriptional control of myelination during mammalian brain development. The authors combine diverse and complementary experimental approaches to generate insights into how DNA looping contributes to transcriptional regulation in functionally specialised cell types. The experiments have been rigorously performed and the main conclusions are justified.

## Introduction

Chromosomes in a single human diploid cell, if linearly stitched together, span a length of more than 2 m. They need to be properly folded to be housed in the cell nucleus with a diameter of 10 μm. Chromosome folding occurs in a dynamic, structured way that regulates gene expression, and DNA replication and repair. Initially discovered as the molecular glue that tethers sister chromatids for

segregation during mitosis (*Haarhuis et al., 2014*; *Uhlmann, 2016*; *Yatskevich et al., 2019*; *Zheng and Yu, 2015*), the cohesin complex has later been shown to be critical for structured chromosome folding and gene expression (*Haarhuis et al., 2017*; *Rao et al., 2017*; *Schwarzer et al., 2017*; *Wutz et al., 2017*).

Cohesin is loaded on chromosomes by the cohesin loader NIPBL. The cohesin–NIPBL complex can extrude DNA loops bidirectionally in an ATP-dependent manner (*Davidson et al., 2019*; *Kim et al., 2019*; *Vian et al., 2018*). The chromatin insulator CTCF has been proposed to block loop extrusion by cohesin, establishing topologically associated domains (TADs) and marking TAD boundaries. Chromatin interactions within each TAD are favored whereas inter-TAD interactions are disfavored. Thus, chromosome loops and TADs shape long-range cis–element interactions, such as promoter–enhancer interactions, thereby regulating transcription.

The vertebrate cohesin complex contains four core subunits: the SMC1–SMC3 heterodimeric ATPase, the kleisin subunit RAD21 that links the ATPase heads, and the HEAT-repeat protein STAG1 or STAG2. STAG1 and STAG2 bind to RAD21 in a mutually exclusive manner and create docking sites for several regulatory proteins, including CTCF (*Hara et al., 2014*; *Li et al., 2020*). STAG1 and STAG2 also interact with DNA and the SMC1–SMC3 hinge domains (*Shi et al., 2020*). STAG1 and STAG2 play redundant roles in sister-chromatid cohesion in cultured human cells, as both need to be simultaneously depleted to produce overt cohesion defects (*Hara et al., 2014*).

Mutations of NIPBL and cohesin subunits, including STAG2, result in human developmental diseases termed cohesinopathies, which affect multiple organs and systems (*Remeseiro et al., 2013b*; *Soardi et al., 2017*). In patients with cohesinopathies, mental retardation and neurological abnormalities caused by brain development defects are common (*Piché et al., 2019*). Dysregulation of gene transcription as a result of reduced cohesin functions has been suggested to underlie these developmental defects (*De Koninck and Losada, 2016*; *Remeseiro et al., 2013a*). In addition, several cohesin genes, including *STAG2*, are frequently mutated in a variety of human cancers (*Martincorena and Campbell, 2015*).

In this study, we deleted *Stag2* specifically in the nervous system in the mouse. The *Stag2* cKO mice exhibited deficient myelination. Loss of STAG2 delayed the maturation of oligodendrocytes and reduced chromosome loops in oligodendrocytes and impaired the transcription of myelination-related genes. Our findings establish the requirement for cohesin in proper gene expression in specific cell types and implicate defective myelination as a potential contributing factor to cohesinopathy.

## Results

### *Stag2* ablation in the nervous system causes growth retardation and neurological defects

*Stag1* is required for mammalian embryonic development (*Remeseiro et al., 2012*), indicating that *Stag2* cannot compensate for the loss of *Stag1*. To examine the physiological functions of *Stag2* in the mouse, we created a *Stag2* 'floxed' mouse line (*Stag2^{f/f}*) by homologous recombination with a template that contained two LoxP sites flanking exon 8 (*Figure 1A, B*) and targeted a critical exon (exon 8) of *Stag2*, which is located on the X chromosome, using CRISPR–Cas9 (*Figure 1—figure supplement 1A*). The *Stag2^{null}* embryos showed severe developmental defects and underwent necrosis by E11.5 days (*Figure 1—figure supplement 1B*). Thus, *Stag2* is required for mouse embryonic development, consistent with a previous report (*De Koninck et al., 2020*). *Stag1* and *Stag2* have nonredundant developmental functions.

To study the functions of STAG2 in adult mice, we crossed the *Stag2^{f/f}* mice with mice bearing the *Rosa26^{CreErt2}* genomic insertion and generated *Stag2^{f/y};Rosa26^{CreErt2}* progenies. The *Stag2^{f/y};Rosa26^{CreErt2}* adult mice were injected with tamoxifen to induce *Stag2* deletion in the whole body (*Figure 1—figure supplement 2A*). Genotyping analysis of blood extracts showed that tamoxifen-induced efficient disruption of the *Stag2* gene locus in *Stag2^{f/y};Rosa26^{CreErt2}* mice (*Figure 1—figure supplement 2B, C*). These *Stag2*-deficient adult mice did not show early onset of spontaneous tumor formation, indicating that *Stag2* mutation alone in somatic cells of mice is insufficient to induce tumorigenesis. The *Stag2*-deficient mice also did not have other obvious adverse phenotypes (*Figure 1—figure supplement 2D*), except that they had slightly lower body weight (*Figure 1—figure supplement 2E, F*), probably due to tissue homeostasis alterations reported by others (*De Koninck et al., 2020*).

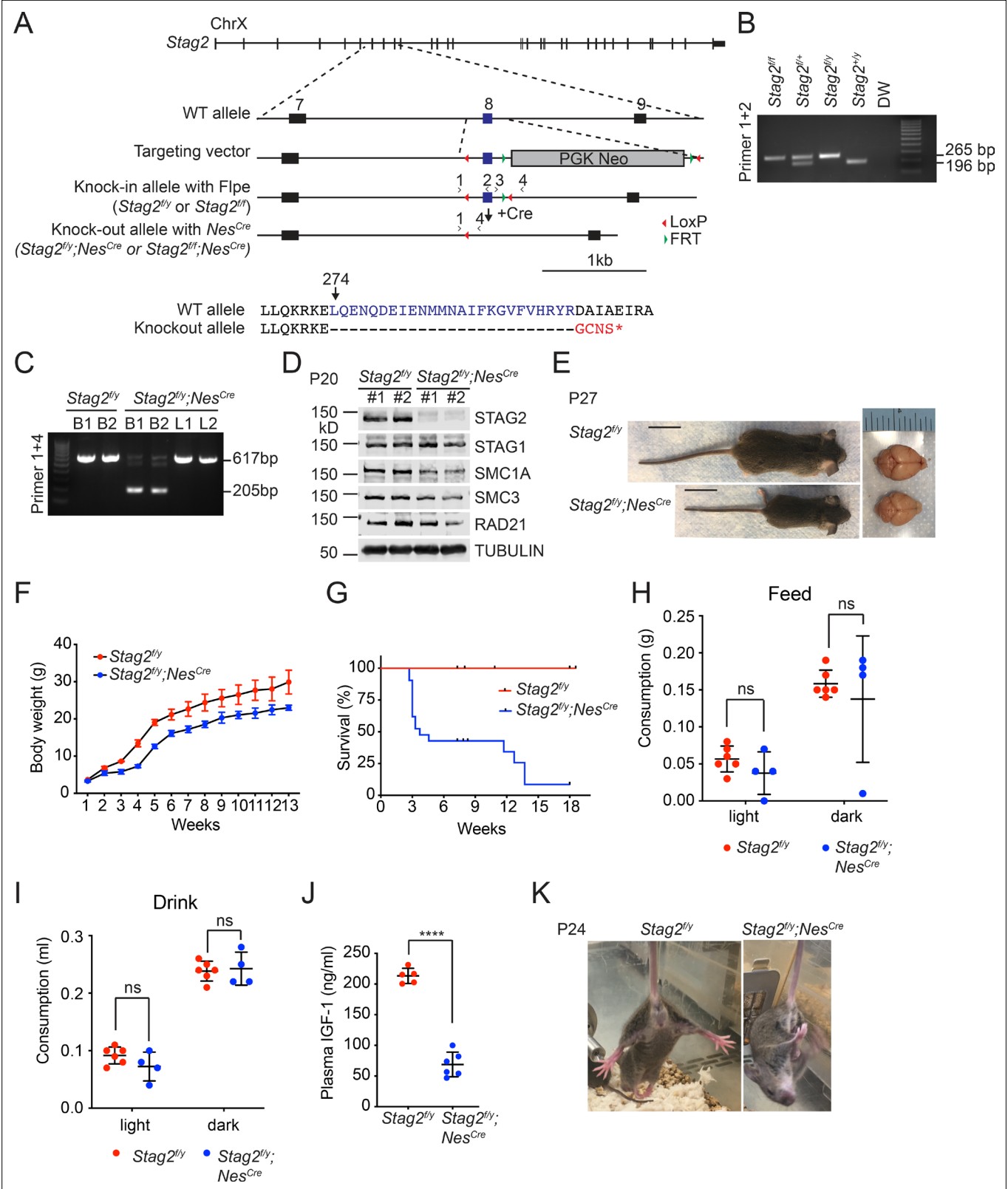

**Figure 1.** *Stag2* ablation in the mouse nervous system causes growth retardation and neurological defects. (**A**) Scheme for creating the 'floxed' *Stag2* allele by gene targeting. The genomic structure of the wild-type (WT) *Stag2* locus, the targeting vector, the knockin allele, the disrupted allele after Cre-mediated recombination, and the positions of the genotyping primers are shown. The amino acid sequence of the knockout allele in the targeted region is shown and aligned with that of the WT allele. (**B**) PCR analysis of the genomic DNA extracted from the tails of indicated mice with the primers

*Figure 1 continued on next page*

Figure 1 continued

in (**A**). (**C**) PCR analysis of genomic DNA extracted from brains (B) or livers (L) of indicated mice. (**D**) Immunoblots of brain lysates of *Stag2^f/y^* and *Stag2^f/y^;Nes^Cre^* mice with antibodies recognizing cohesin subunits and TUBULIN (as the loading control). (**E**) Representative images of *Stag2^f/y^* and *Stag2^f/y^;Nes^Cre^* mice. Scale bar = 2 cm. (**F**) Body weight of *Stag2^f/y^* and *Stag2^f/y^;Nes^Cre^* mice at different age. Mean ± standard deviation (SD) of at least three mice of the same age. (**G**) Survival curves of *Stag2^f/y^* (n = 12) and *Stag2^f/y^;Nes^Cre^* (n = 21) mice. Food (**H**) and water (**I**) consumption of 7- to 8-week-old *Stag2^f/y^* (n = 6) and *Stag2^f/y^;Nes^Cre^* (n = 4) mice. Mean ± SD; ns, not significant. (**J**) Plasma IGF-1 levels of 2-month-old *Stag2^f/y^* (n = 5) and *Stag2^f/y^;Nes^Cre^* (n = 6) mice. Mean ± SD; ****p < 0.0001. (**K**) Representative images of limb-clasping responses of *Stag2^f/y^* and *Stag2^f/y^;Nes^Cre^* mice. The uncropped images of blots in (B–D) are included in ***Figure 1—source data 1***.

The online version of this article includes the following source data and figure supplement(s) for figure 1:

**Source data 1.** Uncropped images of gels and blots in ***Figure 1***.

**Figure supplement 1.** Generation of *Stag2* knockout mice using the CRISPR/Cas9 method.

**Figure supplement 2.** Generation of *Stag2* conditional knockout mice by gene targeting.

**Figure supplement 2—source data 1.** Uncropped images of gels and blots in ***Figure 1—figure supplement 2***.

STAG2 mutations are found in human cohesinopathy patients with mental retardation and neuro-psychiatric behaviors (***Soardi et al., 2017***). To study the function of STAG2 in the nervous system, we generated *Stag2* conditional knockout mice (*Stag2* cKO) by crossing *Stag2^f/f^* mice with Nestin-Cre mice (***Giusti et al., 2014***; ***Figure 1C, D***). The progenies were born in the Mendelian ratio, but *Stag2^f/y^;Nes^Cre^* pups presented growth retardation and premature death (***Figure 1E, G***). More than 50% *Stag2^f/y^;Nes^Cre^* mice died aged about 3 weeks while the rest died at about 4 months. *Stag2^f/y^* mice did not show differences discernible from wild-type (WT) littermates. Although *Stag2^f/y^;Nes^Cre^* mice did not present microcephaly, they exhibited frequent hydrocephaly that might contribute to their premature death. The *Stag2^f/y^;Nes^Cre^* mice displayed normal drinking and feeding behaviors (***Figure 1H***), but showed reduced plasma IGF-1 levels compared to the control mice (***Figure 1J***). *Stag2^f/y^;Nes^Cre^* mice showed forepaw and hindlimb clasping (***Figure 1K***) and limb tremors (***Video 1***), which were not seen in *Stag2^f/y^* mice. These data indicate that *Stag2* deficiency in the nervous system causes growth retardation and neurological defects.

## *Stag2* ablation causes hypomyelination

Hematoxylin and eosin (H&E) staining of brain sections of *Stag2^f/y^;Nes^Cre^* mice did not reveal overt anatomical defects (***Figure 2—figure supplement 1A***). As revealed by immunohistochemistry assays using neuron- or astrocyte-specific antibodies, the differentiation of neurons and astrocytes in *Stag2*-deleted brains was largely normal (***Figure 2—figure supplement 1B–E***). To understand the origins of neurological defects caused by *Stag2* deletion, we analyzed the gene expression changes in *Stag2^f/y^;Nes^Cre^* mouse brains at post-natal day 21 by RNA-sequencing (RNA-seq) (***Figure 2A***). Compared with the control groups, 105 and 62 genes were significantly down- or upregulated by more than twofolds, respectively, in the *Stag2*-deficient brains. The decreased expression of top differentially expressed genes (DEGs) was confirmed by reverse transcription quantitative PCR (RT-qPCR) (***Figure 2B***). Among the 105 downregulated DEGs in the brains of *Stag2* cKO mice, 44 were enriched in myelin (***Figure 2C***; ***Thakurela et al., 2016***). The ingenuity pathway analysis (IPA) pinpoints cholesterol biosynthesis pathways as the most affected canonical pathways (***Figure 2D*** and ***Supplementary file 1***). We further confirmed that the cholesterol biosynthesis precursors were reduced in *Stag2^f/y^;Nes^Cre^* brains (***Figure 2—figure supplement 1F***).

Myelin is the membrane sheath that wraps around axons to facilitate rapid nerve conduction and maintain metabolic supply (***Williamson and Lyons, 2018***). Dynamic myelination in the central nervous system (CNS) is critical for proper neurodevelopment, and defective myelination is associated with autoimmune and neurodegenerative diseases (***Mathys et al., 2019***; ***Wolf et al., 2021***). Cholesterol biosynthesis is essential for normal myelination (***Hubler et al., 2018***; ***Saher et al.,***

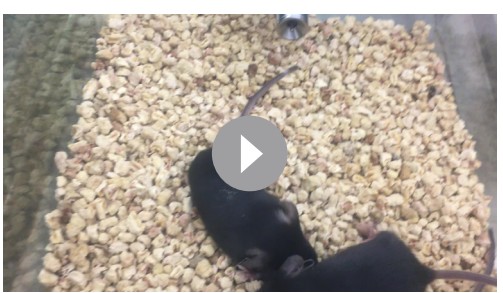

**Video 1.** Neurological defects of brain-specific *Stag2* KO mice.

https://elifesciences.org/articles/77848/figures#video1

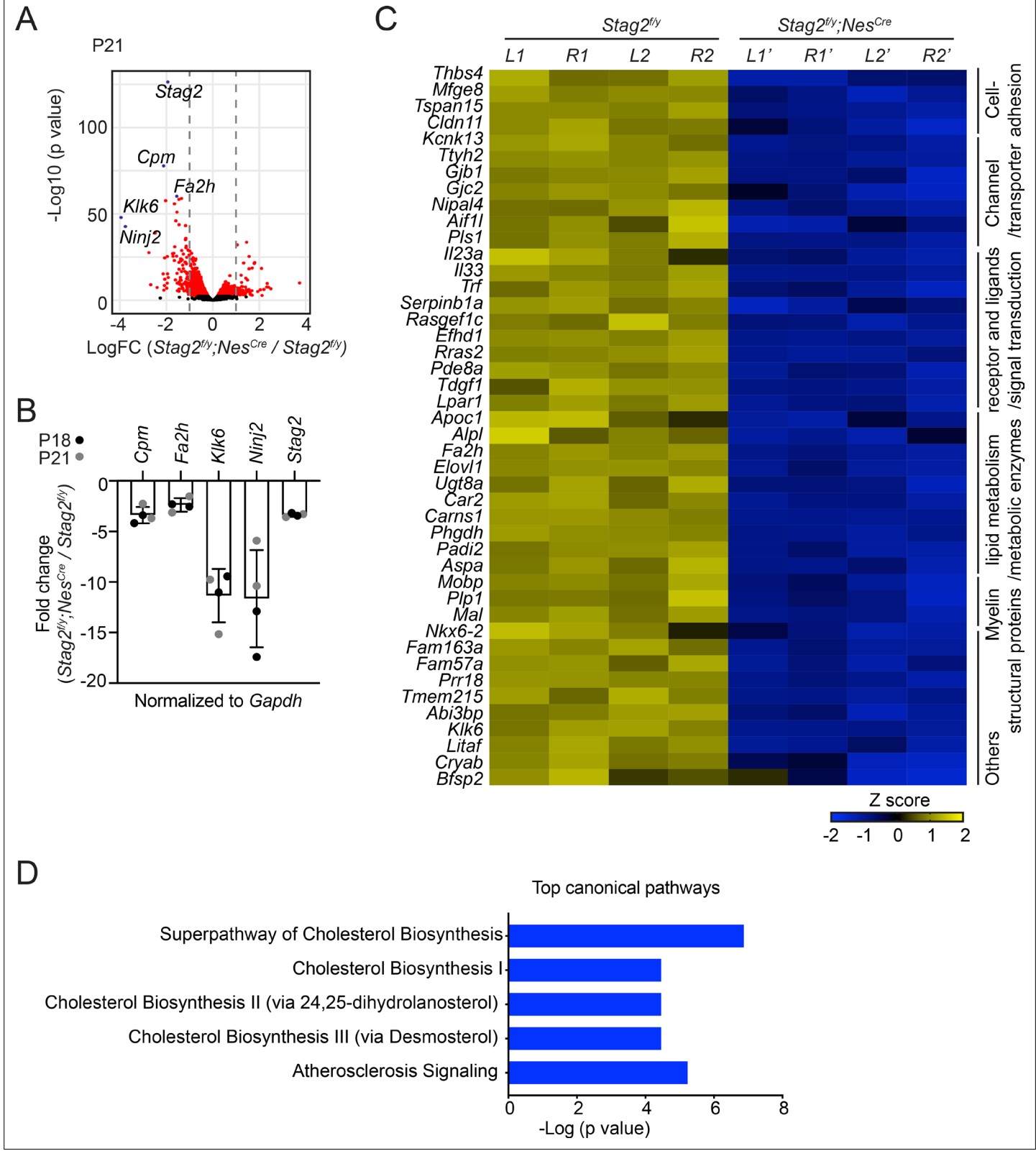

**Figure 2.** *Stag2* ablation in mouse brains downregulates the expression of myelin genes. (**A**) Volcano plot of bulk RNA-sequencing results of *Stag2^f/y* and *Stag2^f/y;Nes^Cre* brain extracts. Top differentially expressed genes (DEGs) are colored blue and labeled. *n* = 4 pairs of P21 *Stag2^f/y* and *Stag2^f/y;Nes^Cre* brain hemispheres were used for the comparison. (**B**) Reverse transcription quantitative PCR (RT-qPCR) analysis of the top downregulated genes in the brain extracts. *n* = 4 pairs of *Stag2^f/y* and *Stag2^f/y;Nes^Cre* littermates were used. Mean ± standard deviation (SD). (**C**) Heatmap of the expression of myelin-

*Figure 2 continued on next page*

*Figure 2 continued*

enriched genes that were downregulated by more than twofolds in *Stag2^{f/y}*;*Nes^{Cre}* brains. *L1* and *R1*, left and right brain hemispheres of the *Stag2^{f/y}#1* mouse. *L2* and *R2*, left and right brain hemispheres of the *Stag2^{f/y}#2* mouse. *L1'* and *R1'*, left and right brain hemispheres of the *Stag2^{f/y}*;*Nes^{Cre}* #1 mouse. *L2'* and *R2'*, left and right brain hemispheres of the *Stag2^{f/y}*;*Nes^{Cre}#2* mouse. The biological pathways of these genes are labeled on the right. (**D**) The top 5 canonical pathways identified by ingenuity pathway analysis (IPA) of the DEGs. The complete gene list is used as the background.

The online version of this article includes the following figure supplement(s) for figure 2:

**Figure supplement 1.** STAG2 deficiency in mouse brains attenuates cholesterol biosynthesis.

**Figure supplement 2.** Over-representation analysis (ORA) of the RNA-sequencing (RNA-seq) results of the mouse brain samples.

*2005*). Ensheathment of neurons and gliogenesis were among the top enriched biological pathways in downregulated DEGs (*Figure 2—figure supplement 2*). The innate immune response was among the top enriched pathways in the upregulated DEGs. We hypothesized that depletion of STAG2 caused myelination defects in the nervous system.

Indeed, brain sections of *Stag2^{f/y}*;*Nes^{Cre}* mice showed greatly reduced luxol fast blue (LFB) staining compared to those of *Stag2^{f/y}* and *Nes^{Cre}* heterozygous mice at about P21 (*Figure 3A* and *Figure 3—figure supplement 1A*). Immunohistochemistry using antibodies against myelin proteins, Myelin basic protein (MBP) and Proteolipid protein 1 (PLP1), confirmed that *Stag2* cKO mice had significant defects in myelin fiber formation at P18-P21 (*Figure 3B–F*). In both cerebral cortex and cerebellum, there were fewer and sparser myelin fibers in *Stag2^{f/y}*;*Nes^{Cre}* mice, as compared to the *Stag2^{f/y}* controls. Axon myelin ensheathment was further examined using transmission electron microscopy (*Figure 3G*). *Stag2^{f/y}*;*Nes^{Cre}* mice at P18 had significantly fewer myelin-wrapped axons at optic nerves. Collectively, these data indicate insufficient myelination in the *Stag2* cKO mice. Myelination predominantly occurs at 3 weeks after birth in the mouse. The timing of premature death of *Stag2* cKO mice is thus consistent with defective myelination as a contributing factor to the lethality.

We examined *Stag1* and *Stag2* expression patterns in P18 WT mouse brains by in situ hybridization using isotope-labeled RNA probes (*Figure 3—figure supplement 1B*). Both *Stag1* and *Stag2* were expressed at high levels in hippocampus, medial habenula, neocortex, and cerebellum granular layer. Aside from these regions, the *Stag2* transcripts were detected at relatively high levels in subventricular zone, thalamus, fiber tracts, midbrain, and hindbrain regions. *Stag2* is thus ubiquitously expressed in the brain.

## STAG2 regulates transcription in OLs

Oligodendrocytes (OLs) are responsible for myelination in the CNS. To examine whether the OL lineage was affected by *Stag2* deletion, we performed single-cell RNA-sequencing (scRNA-seq) analysis of *Stag2^{f/y}*;*Nes^{Cre}* and *Stag2^{f/y}* forebrains at P13. As revealed by clusters in the t-SNE plot, the two genotype groups had similar cellular compositions (*Figure 4A, B*). All cell clusters were present in *Stag2^{f/y}*;*Nes^{Cre}* brains, again indicating generally normal neural cell differentiation. Cell-type identities were discovered with feature gene expression (*Figure 4—figure supplement 1A*). Based on the expression changes of *Stag2* and other cohesin genes in OLs, astrocytes, and neuronal lineages (*Figure 4—figure supplement 1B–D*), it is clear that *Stag2* ablation occurred early in the NPC stage and was maintained in all differentiated cell lineages.

The OL lineage consisted of five clusters: cycling OL progenitors (OPCcycs), OL progenitors (OPCs), newly formed OLs (NFOLs), myelin-forming OLs (mFOLs), and fully matured OLs (MOLs). Quantification of the distributions of these five cell types within the OL lineage revealed a mild reduction in the proportion of MOLs in *Stag2^{f/y}* forebrains (*Figure 4C*). We noticed that a higher percentage of neurons was recovered in the *Stag2^{f/y}*;*Nes^{Cre}* group. Since the bulk RNA-seq results did not show global upregulation of neuron-specific genes, we suspect that neurons in *Stag2^{f/y}*;*Nes^{Cre}* had fewer myelin-wrapped axons and were easier to be dissociated and kept alive during our library preparation for scRNA-seq. Thus, from the transcriptome analysis, we did not observe overt defects in most neural cell differentiation in the *Stag2*-deficient forebrain regions.

We then performed trajectory inference and pseudotime analysis of the OL lineage (*Figure 4—figure supplement 2A, B*). Consistent with our cell-type assignment, pseudotime variables indicated continuous differentiation from OPCs to NFOLs, mFOLs, and MOLs (*Figure 4—figure supplement 2C,D*). The reclustering of single cells in the OL lineage along the pseudotime path revealed that

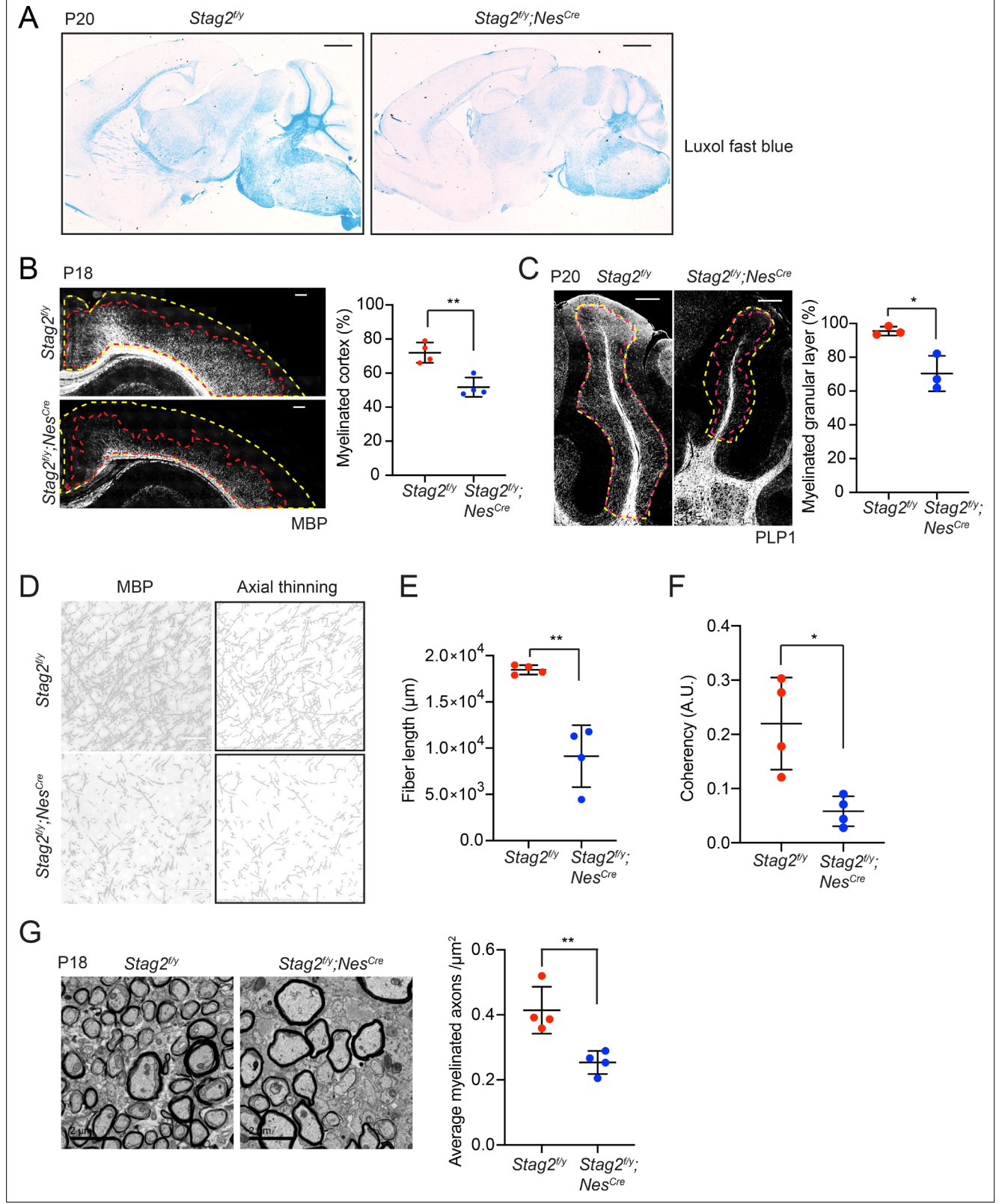

**Figure 3.** *Stag2* ablation in the nervous system compromises myelination during early postnatal development. (**A**) Luxol fast blue staining of the sagittal sections of *Stag2^{f/y}* and *Stag2^{f/y};Nes^{Cre}* brains. *n* = 3 animals per genotype. Scale bar = 1 mm. (**B**) Immunohistochemistry staining with the anti-MBP antibody in the cerebral cortex (left panel). Antibody-stained areas and DAPI staining regions are marked with red and yellow dashed lines, respectively. Scale bar = 200 μm. Quantification of the percentage of the myelinated cortex is shown in the right panel. *n* = 4 pairs of *Stag2^{f/y}* and *Stag2^{f/y};Nes^{Cre}*

*Figure 3 continued on next page*

*Figure 3 continued*

littermates were used (P18 or P21) for the comparison. **p < 0.01; mean ± standard deviation (SD). (**C**) Immunohistochemistry staining with the anti-PLP1 antibody in the cerebellum (left panel). Antibody-stained areas and DAPI staining regions are marked with red and yellow dashed lines, respectively. Scale bar = 200 μm. Quantification of the percentage of the myelinated cerebellum granular layer is shown in the right panel. *n* = 3 pairs of *Stag2^f/y* and *Stag2^f/y;Nes^Cre* littermates were used (P20 or P25) for the comparison. *p < 0.05; mean ± SD. (**D**) Higher magnification images (left panel) of the immunohistochemistry staining with the anti-MBP antibody in (**B**). Images processed through axial thinning are shown in the right panel. Scale bar = 50 μm. Total fiber length (**E**) and fiber coherency (**F**) measured using the processed images in (**D**). *n* = 4 pairs of *Stag2^f/y* and *Stag2^f/y;Nes^Cre* littermates were used (P18 or P21). *p < 0.05, **p < 0.01; mean ± SD. (**G**) Transmission electron microscopy images of the optic nerves (left panel). Scale bar = 2 μm. Quantification of myelinated axon distributions is shown in the right panel. *n* = 4 pairs of P18 *Stag2^f/y* and *Stag2^f/y;Nes^Cre* littermates were used. *n* ≥ 10 fields of each mouse were taken, and the average distribution of myelinated axons were calculated for each mouse and plotted. **p < 0.01; mean ± SD.

The online version of this article includes the following figure supplement(s) for figure 3:

**Figure supplement 1.** Brain-specific *Stag2* deletion impairs central nervous system (CNS) myelination.

more cells were present in the terminal maturation stages in the *Stag2^f/y* brains (***Figure 4—figure supplement 2E, F***). Conversely, more cells were retained at the undifferentiated stages in the *Stag-2^f/y;Nes^Cre* brains. Strikingly, some myelination genes, including Myelin and lymphocyte protein (*Mal*), were specifically repressed in *Stag2^f/y;Nes^Cre* MOLs, with their expression in nonneural cells unaltered (***Figure 4D*** and ***Figure 4—figure supplement 3A***). These observations suggest that STAG2 deficiency delays the maturation of OLs and compromises myelination-specific gene expression in mature OLs. Interestingly, compared to *Stag2* and genes encoding other cohesin core subunits, *Stag1* transcripts are less abundant in the OL lineage, except for cycling OPCs (***Figure 4—figure supplement 3B, C***). The low expression of *Stag1* in mature OLs might make these cells more dependent on *Stag2* for function.

To confirm the transcriptional defects in the OL lineage caused by *Stag2* deletion, we isolated primary OLs at intermediate differentiation stages from *Stag2^f/y;Nes^Cre* and *Stag2^f/y* forebrains at P12-P14 with antibody-conjugated magnetic beads and conducted bulk RNA-seq analysis (***Figure 4E***). For both genotypes, the marker genes for NFOL and mFOLs were highly expressed in the isolated primary OLs (***Figure 4—figure supplement 4A***), suggesting that they mainly contained these two cell types. In *Stag2*-deleted OLs, 271 and 292 genes were downregulated or upregulated by more than two folds, respectively (***Figure 4F*** and ***Supplementary file 2***). Intriguingly, the downregulated genes were generally highly expressed in WT cells, whereas the upregulated genes had low expression levels in WT cells (***Figure 4—figure supplement 4B–D***). The top pathways enriched in the downregulated DEGs included the cholesterol and small molecule biosynthetic pathways and oligodendrocyte differentiation (***Figure 4G*** and ***Figure 4—figure supplement 5***). Cilium organization and assembly are the top enriched pathways in the upregulated DEGs (***Figure 4—figure supplement 6***). Among the 105 downregulated DEGs identified by RNA-seq analysis of the whole brain of *Stag2*-deficient mice, 42 were also differentially expressed in primary oligodendrocytes (***Figure 4H***). The cholesterol biosynthetic pathways were recognized as the major altered pathways (***Figure 4—figure supplement 4E***). Thus, defective cholesterol biosynthesis and oligodendrocyte differentiation likely underly hypomyelination and neurological defects in *Stag2* cKO mice.

We performed chromatin immunoprecipitation sequencing (ChIP-seq) experiments to examine the enrichment of the active transcription mark H3K27ac in *Stag2^f/y* and *Stag2^f/y;Nes^Cre* OLs and found that *Stag2* loss did not appreciably affect H3K27Ac enrichment at transcription start sites (TSSs) (***Figure 5A, B***). Consistent with our RNA-seq results, the upregulated genes had much lower H3K27ac enrichment near their TSS, indicating that they were less active. We then checked the genomic distribution of STAG2 by ChIP-seq. Among other genomic loci, STAG2 was enriched at TSS of stable and downregulated genes, including genes in the cholesterol biosynthesis and myelination pathways (***Figure 5C, D***, ***Figure 5—figure supplement 1***, and ***Supplementary file 2***). Among the 271 downregulated DEGs, there were 210 genes (77%) with STAG2 enrichment near the transcriptional start site (TSS ± 2 kb). Thus, STAG2 occupied the promoter regions of many downregulated DEGs in oligodendrocytes. It was enriched at the TSS of upregulated genes to a lesser extent, with only 117 of the 292 (40%) upregulated DEGs exhibiting STAG2 ChIP-seq peaks at their TSS ± 2 kb regions. *Stag2* loss might have indirectly affected the expression of these less active genes.

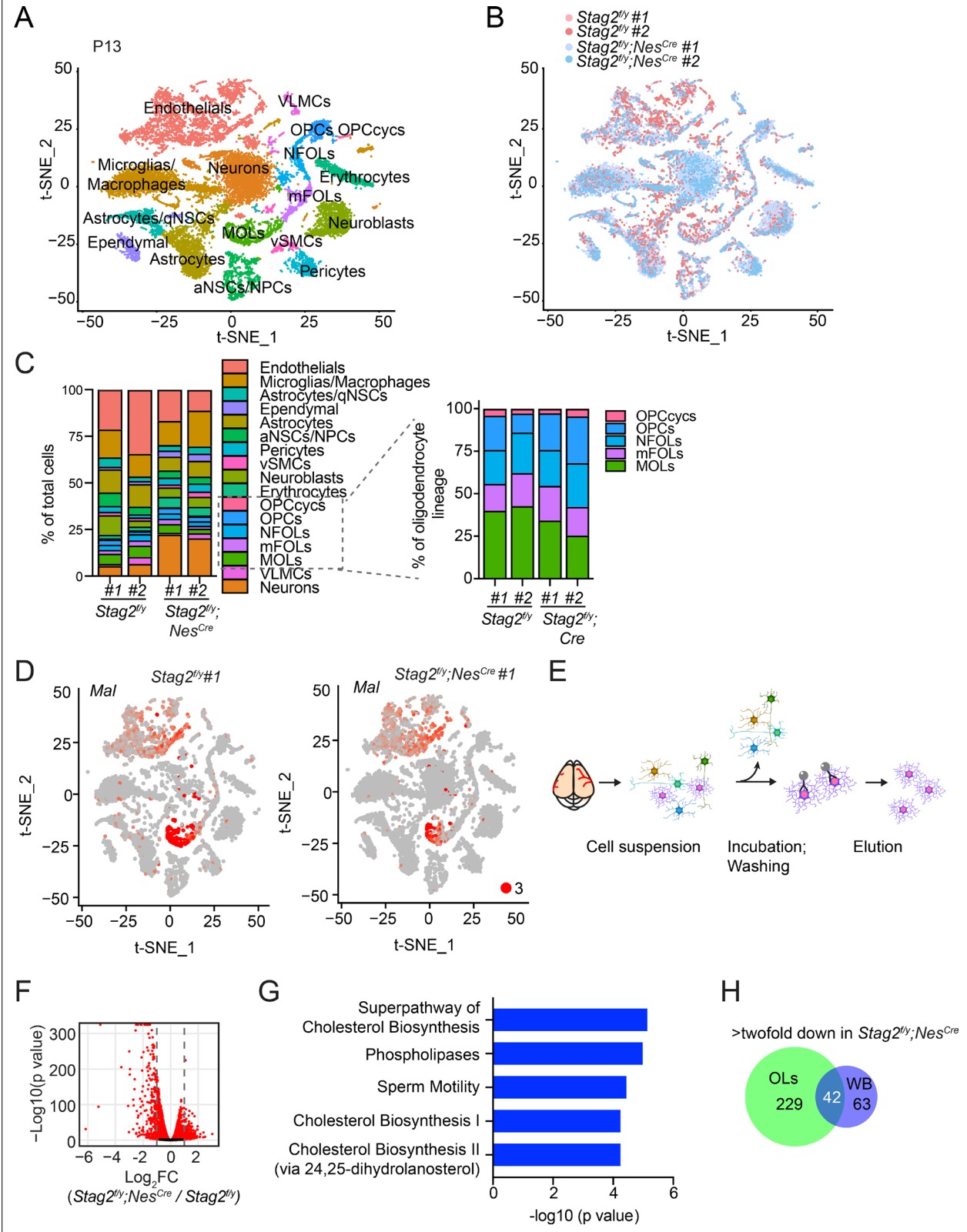

**Figure 4.** Deletion of *Stag2* in mouse brains causes differentiation delay and transcriptional changes in oligodendrocytes. (**A**) *t*-SNE plot of cell clusters in *Stag2^{f/y}* and *Stag2^{f/y};Nes^{Cre}* forebrains analyzed by single-cell RNA-sequencing (scRNA-seq). *n* = 2 mice of each genotype were used in the scRNA-seq analysis. aNSCs/NPCs, active neural stem cells or neural progenitor cells; Astrocytes/qNSCs, astrocytes or quiescent neural stem cells; OPCcycs, cycling oligodendrocyte (OL) progenitor cells; OPCs, OL progenitor cells; NFOLs, newly formed OLs; mFOLs, myelin-forming OLs; MOLs, matured OLs;

*Figure 4 continued on next page*

*Figure 4 continued*

VLMCs, vascular and leptomeningeal cells; vSMCs, vascular smooth muscle cells. (**B**) *t*-SNE clustering as in (**A**) but colored by genotype. (**C**) Left panel: cell-type composition and percentage as colored in (**A**). Right panel: percentage of cell clusters of the oligodendrocyte lineage. (**D**) FeaturePlot of a representative gene (*Mal*) specifically suppressed in MOLs of *Stag2^{f/y};Nes^{Cre}* forebrains. A maximum cutoff of 3 was used. (**E**) Experimental scheme of the magnetic-activated cell sorting (MACS) of primary OLs. (**F**) Volcano plot of bulk RNA-sequencing (RNA-seq) results of *Stag2^{f/y}* and *Stag2^{f/y};Nes^{Cre}* primary OLs. (**G**) The top 5 canonical pathways identified by ingenuity pathway analysis (IPA) of the differentially expressed genes (DEGs) with more than twofold change in (**F**). The complete gene list is used as the background. (**H**) Commons DEGs shared between bulk RNA-seq analyses of the whole brains (WB) and primary OLs.

The online version of this article includes the following figure supplement(s) for figure 4:

**Figure supplement 1.** *Stag2* is ablated during early neural lineage differentiation of *Stag2* knockout mice.

**Figure supplement 2.** *Stag2* deletion causes differentiation delay in the oligodendrocyte lineage.

**Figure supplement 3.** STAG2 regulates the transcription of oligodendrocyte genes.

**Figure supplement 4.** STAG2 regulates transcription in primary oligodendrocytes.

**Figure supplement 5.** Over-representation analysis (ORA) of the downregulated genes in *Stag2*-deleted oligodendrocytes.

**Figure supplement 6.** Over-representation analysis (ORA) of the upregulated genes in *Stag2*-deleted oligodendrocytes.

### *Stag2* deletion does not alter compartments or TADs in OLs

To investigate whether chromosome conformation was altered by *Stag2* deletion and whether that caused transcription dysregulation, we performed high-dimensional chromosome conformational capture (Hi-C) analysis of primary OLs isolated from *Stag2^{f/y}* and *Stag2^{f/y};Nes^{Cre}* mice in biological replicates (*Figure 6* and *Figure 6—figure supplement 1*). We observed few compartment switching events in *Stag2*-deleted cells (*Figure 6A–C*). Virtually all genomic regions in *Stag2*-deleted cells were kept in their original compartment categories (AA or BB) (*Figure 6C*). Only a very small number of genomic regions switched compartments (AB or BA). Consistent with the RNA-seq data, analysis of average gene expression changes of DEGs in these genomic regions revealed that more genes located in the transcriptionally active A compartment (AA) were repressed in *Stag2*-deleted cells and more genes in the transcriptionally silent B compartment (BB) were upregulated (*Figure 6D* and *Figure 6—figure supplement 1C*). Genes that switched from the A compartment to the B compartment were not more repressed compared to those that remained in the A compartment. Likewise, compared to genes that stayed in the B compartment, genes located in chromatin regions that switched from compartment B to A were not significantly activated. Acute depletion of all forms of cohesin eliminates TAD formation (*Wutz et al., 2017*). In contrast, deletion of *Stag2* had minimal impact on TAD formation in oligodendrocytes (*Figure 6E–G* and *Figure 6—figure supplement 1D*), suggesting that STAG1-cohesin compensates for the loss of STAG2-cohesin in spatial organization of chromatin at larger than megabase scales. Therefore, our analyses did not uncover evidence for compartment switching and TAD alterations being the underlying cause for the observed gene expression changes in STAG2-deficient OLs.

### Promoter-anchored loops were reduced in *Stag2*-deleted OLs

While TAD boundaries are largely conserved among species and cell types, chromatin interactions within each TAD are more flexible and variable in cells undergoing differentiation, tumorigenesis, and reprogramming (*Dixon et al., 2015*; *Dixon et al., 2012*). Among the intra-TAD chromatin interactions, the enhancer–promoter loops are particularly important for transcription and are often cell-type specific. We examined whether chromatin loops in OLs were affected by *Stag2* loss. Compared to *Stag2^{f/y}* OLs, *Stag2^{f/y};Nes^{Cre}* OLs had significantly fewer loops across almost all genomic distances (*Figure 7A, B* and *Figure 7—figure supplement 1*). The common and genotype-specific loops are reproducible in each replicate. Loops specific to *Stag2^{f/y};Nes^{Cre}* OLs, which were likely mediated by STAG1-cohesin, were longer than STAG2-dependent *Stag2^{f/y}*-specific loops. When genomic distances exceeded 0.25 Mb, the loops from *Stag2^{f/y};Nes^{Cre}* cells gradually gained higher scores over loops from *Stag2^{f/y}* cells (*Figure 7C*). Therefore, STAG1-cohesin cannot completely compensate for STAG2-cohesin during loop formation. STAG1-cohesin-mediated loops are relatively longer than STAG2-cohesin-mediated loops, consistent with published findings in HeLa cells (*Wutz et al., 2020*).

We then tested whether the loop number decrease in *Stag2*-deficient cells could be a cause for transcriptional changes. When examining the local Hi-C maps, we noticed that loops anchored at

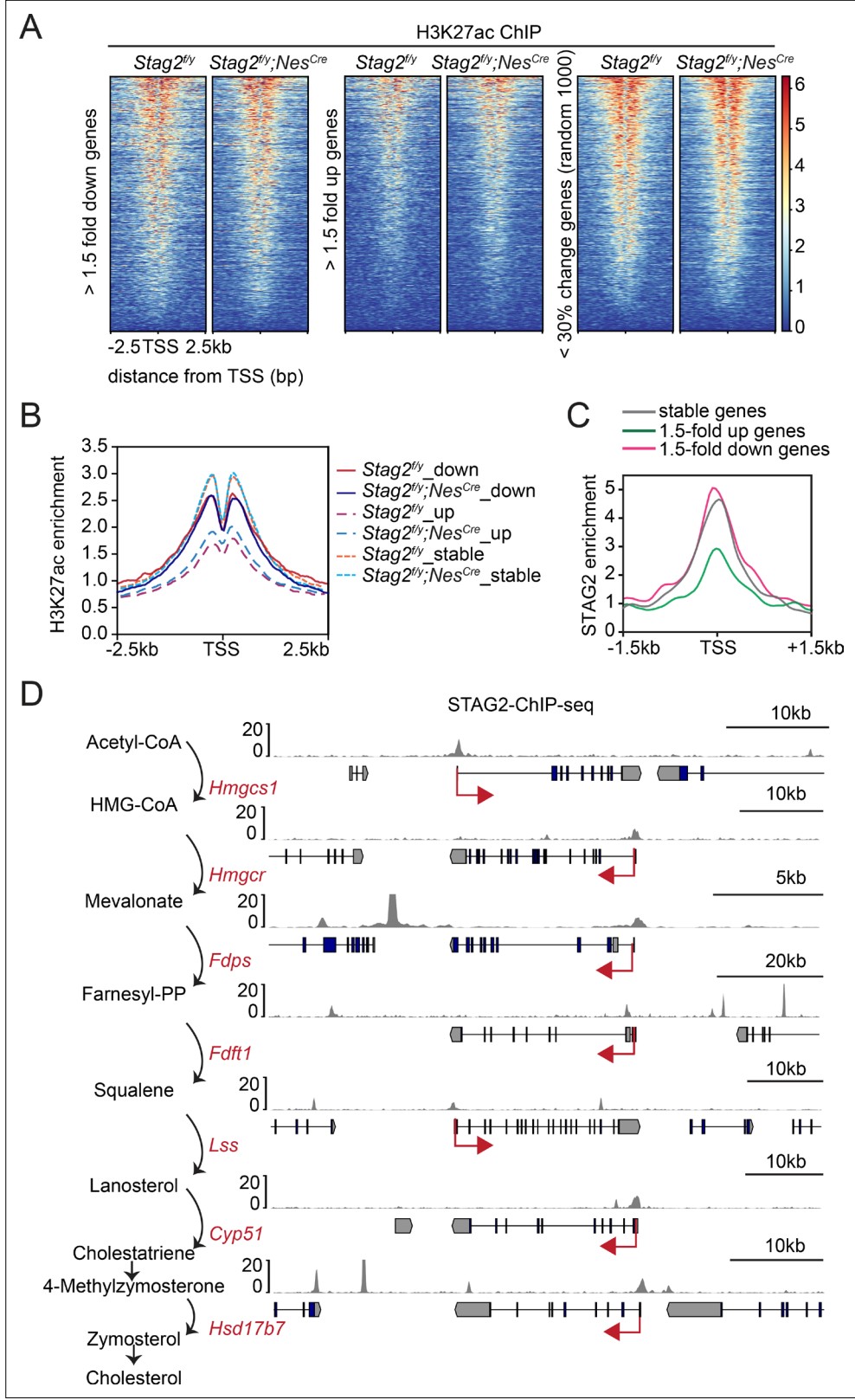

**Figure 5.** Enrichment of STAG2 and histone modifications at gene promoters. (**A**) Heatmap of H3K27ac chromatin immunoprecipitation sequencing (ChIP-seq) signal enrichment in the promoter regions of genes in the indicated categories. (**B**) Density profile of H3K27ac ChIP-seq signal enrichment in the promoter regions of genes in the indicated categories as in (**A**). (**C**) Density profile of STAG2 ChIP-seq signal enrichment in the promoter regions of

*Figure 5 continued*

genes in the indicated categories as in (**A**). (**D**) Binding of STAG2 at the genomic loci of downregulated genes that encode cholesterol biosynthetic enzymes as revealed by ChIP-seq.

The online version of this article includes the following figure supplement(s) for figure 5:

**Figure supplement 1.** STAG2 occupies the promoters of myelination genes.

gene promoters, including those of downregulated genes, were reduced in *Stag2^{f/y}*;*Nes^{Cre}* oligodendrocytes (*Figure 7D*, *Figure 7—figure supplement 2A–C*, and *Supplementary file 2*). The effects were again reproducible in each replicate (*Figure 7—figure supplement 2D*). Promoter-anchored loops (P-loops) can potentially be promoter–promoter links, promoter–enhancer links, and gene loops. The total number of P-loops was proportionally decreased in *Stag2^{f/y}*;*Nes^{Cre}* cells (*Figure 7—figure supplement 3A*). Moreover, the loops anchored at the downregulated genes were stronger than those at upregulated and stable genes (*Figure 7—figure supplement 3B*). We then compared P-loops associated with DEGs using pile-up analysis of local contact maps. Loop enrichment at promoters of downregulated genes was reduced in *Stag2^{f/y}*;*Nes^{Cre}* cells to a greater extent than that at promoters of upregulated and stable genes (*Figure 7E*). Among the 162 downregulated DEGs with reduced promoter-anchored loops in the *Stag2*-depleted cells, 137 genes (85%) had STAG2 peaks in their promoter regions (TSS ± 2 kb). The loops anchored at downregulated genes with STAG2 binding had significantly higher loop scores, compared to those with no STAG2 binding (*Figure 7—figure supplement 3C, D*). This difference was still observed in *Stag2*-deleted cells, suggesting that the stronger looping at these gene promoters might be maintained by STAG1-cohesin or other factors in these cells. By contrast, the loops anchored at upregulated genes with STAG2 binding had lower loop scores. These differences became insignificant in the *Stag2*-deleted cells. The loop scores of loops anchored at stable genes were not affected by STAG2 occupancy. Taken together, our results suggest that *Stag2* loss diminishes the number of, but not the strength, short chromosome loops, including promoter-anchored loops. Highly expressed genes might be more reliant on these loops for transcription and are preferentially downregulated by *Stag2* loss (*Figure 4—figure supplement 4B-D*).

We also performed pile-up analysis of local chromatin regions flanking TSSs (*Figure 7F*). Strikingly, we observed a clear stripe that extended from the TSS of downregulated gene only in the direction of transcription. The formation of promoter-anchored stripes (P-stripes) on aggregated plots is consistent with one-sided loop extrusion from the promoter to the gene body. The P-stripe was still present in *Stag2^{f/y}*;*Nes^{Cre}* cells, suggesting that STAG1 could compensate for the loss of STAG2 and mediate its formation (*Figure 7F* and *Figure 7—figure supplement 3E*).

## Discussion

Cohesin is critical for the three-dimensional (3D) organization of the genome by extruding chromosome loops. Acute depletion of cohesin abolishes chromosome loops and TADs, but has moderate effects on transcription. The two forms of cohesin in vertebrate somatic cells, namely STAG1-cohesin and STAG2-cohesin, have largely redundant functions in supporting sister-chromatid cohesion and cell viability, but they have nonredundant functions during development. In this study, we have established a myelination-promoting function of STAG2 in the CNS in the mouse. We further provide evidence linking hypomyelination caused by STAG2 loss to reduced promoter-anchored loops at myelination genes in oligodendrocytes.

### Myelination functions of STAG2 and implications for cohesinopathy

Selective ablation of *Stag2* in the nervous system in the mouse causes growth retardation, neurological defects, and premature death. STAG2 loss delays the maturation of oligodendrocytes and reduces the expression of highly active myelin and cholesterol biosynthesis genes in oligodendrocytes, resulting in hypomyelination in the CNS. Hypomyelination disorders in humans and mice are known to produce abnormal neurological behaviors similar to those seen in our *Stag2* cKO mice, suggesting that hypomyelination is a major underlying cause for the phenotypes in *Stag2* cKO mice. The growth retardation in these mice can be explained by insufficient secretion of growth hormones, which may be a consequence of defective neuronal signaling.

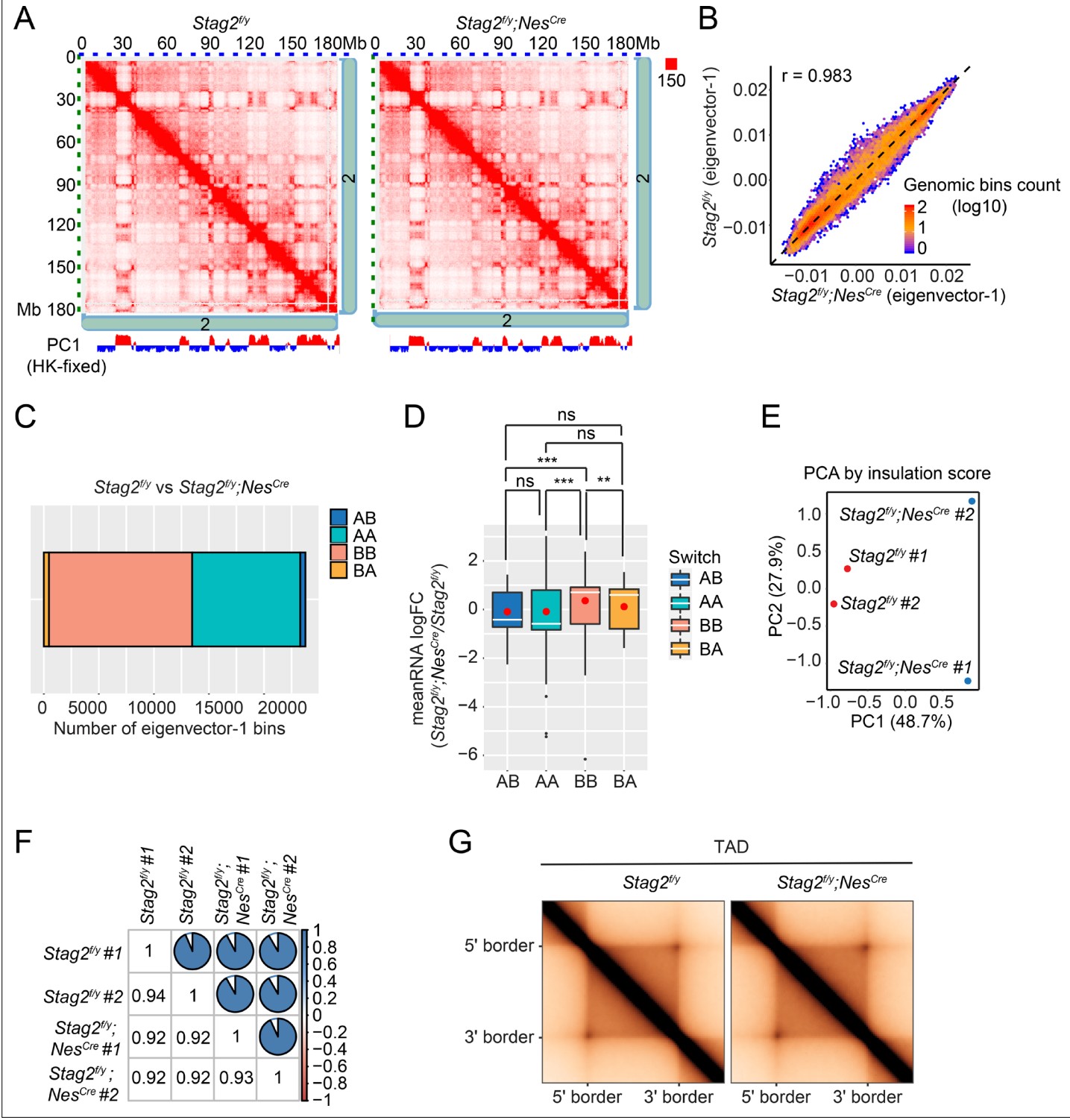

**Figure 6.** Loss of *Stag2* does not alter compartments and topologically associated domains (TADs) in oligodendrocytes. (**A**) Representative snapshots of balanced Hi-C contact matrices of chromosome 2. Tracks of eigenvector-1 fixed with housekeeping genes are shown below, with A and B compartments shown in red and blue, respectively. (**B**) Hexbin plot of eigenvector-1 for genomic bins (100 kb) in *Stag2^f/y^* and *Stag2^f/y^;Nes^Cre^* oligodendrocytes (OLs). (**C**) Chromatin bins were classified into four categories based on the eigenvector sign and whether it has switched with a delta bigger than 1.5. AB, changing from compartment A in *Stag2^f/y^* to compartment B in *Stag2^f/y^;Nes^Cre^*; BA, from B in *Stag2^f/y^* to A in *Stag2^f/y^;Nes^Cre^*; AA, A in both *Stag2^f/y^* and *Stag2^f/y^;Nes^Cre^*; BB, B in both *Stag2^f/y^* and *Stag2^f/y^;Nes^Cre^*. (**D**) Boxplot of averaged gene expression change of differentially expressed genes (DEGs) (RNA logFC cutoff of ±0.58) inside each genomic bin. Bins counted: AA, 1646; AB, 56; BA, 69; BB, 910. Red dots represent the mean value. An unpaired

*Figure 6 continued on next page*

*Figure 6 continued*

Wilcoxon test was used for the statistical analysis. **p < 0.01; ***p < 0.001; ns, not significant. Principal component (**E**) and similarity (**F**) analysis performed using the insulation score at 10 kb resolution. (**G**) Aggregate TAD analysis on the 10 kb merged Hi-C matrices using TADs called from the merged samples of *Stag2^(f/y)* at 10 kb resolution.

The online version of this article includes the following figure supplement(s) for figure 6:

**Figure supplement 1.** *Stag2*-deleted OLs do not present significant changes in compartments and topologically associated domains (TADs).

Mutations of cohesin subunits and regulators, including STAG2, cause the Cornelia de Lange syndrome (CdLS) and other similar developmental diseases, collectively termed cohesinopathy. CdLS patients exhibit short stature and developmental defects in multiple tissues and organs, including the brain. Although STAG2 mutations are implicated in human cohesinopathy, these mutations are rare and hypomorphic (*Soardi et al., 2017*). The cohesin loader NIPBL is the most frequently mutated cohesin regulator in cohesinopathy (*Mannini et al., 2013*). NIPBL deficiency is expected to affect the functions of both STAG1- and STAG2-cohesin. It is possible that the partial loss of STAG2-cohesin function leads to subtle myelination defects in patients with cohesinopathy. Indeed, lack of myelination in certain brain regions of CdLS patients has been reported (*Avagliano et al., 2017*; *Vuilleumier et al., 2002*). As myelination of the CNS mostly occurs after birth and during childhood, strategies aimed at enhancing myelination might help to alleviate certain disease phenotypes and symptoms.

## Mechanisms by which STAG2 promotes myelination

STAG2 promotes oligodendrocyte maturation and the expression of myelination genes in mature oligodendrocytes. Because STAG2 does not have an established cohesin-independent function, it most likely activates the myelination-promoting transcriptional program as a core component of cohesin. Consistent with previous reports (*Rao et al., 2017*), loss of STAG2-cohesin in oligodendrocytes does not affect genome compartmentalization, but reduces the number of relatively short chromosome loops, including promoter-anchored loops. Promoter-anchored loops at downregulated genes are reduced to a greater extent than those at stable and upregulated genes. These findings suggest that STAG2-cohesin promotes the myelination transcriptional program by forming promoter-anchored loops.

Pile-up analysis of Hi-C maps reveals the formation of asymmetric promoter-anchored stripes in the direction of transcription at downregulated genes, indicative of active loading of cohesin at TSSs followed by one-sided loop extrusion from the promoter to the gene body. The stripes are, however, not reduced in STAG2-deficient cells. Because both forms of cohesin are capable of loop extrusion, it is possible that STAG1-cohesin can compensate for the loss of STAG2-cohesin in loop extrusion. It remains to be tested whether the intrinsic kinetics and processivity of loop extrusion mediated by the two forms of cohesin are differentially regulated by cellular factors or posttranslational modifications and whether these differences contribute to their nonredundant roles in transcription regulation.

We envision three possibilities that may account for why oligodendrocytes, but not other cell types, are more severely affected by *Stag2* loss in the CNS. First, STAG2-cohesin may be more abundant than STAG1-cohesin in postmitotic OLs, making them more dependent on STAG2 for proper functions. Second, STAG1-cohesin preferentially localizes to CTCF-enriched TAD boundaries whereas STAG2-cohesin is more enriched at enhancers lacking CTCF (*Kojic et al., 2018*). Enhancers are critical for cell-type-specific gene transcriptional programs. To cooperate with the axonal growth during postnatal neurodevelopment, enhancer-enriched transcription factors induce timely and robust gene expression in oligodendrocytes for proper myelination (*Mitew et al., 2014*). The high demand for enhancer function may render the transcription of myelination genes more reliant on STAG2-cohesin. Finally, the C-terminal regions of STAG1 and STAG2 are divergent in sequence and may bind to different interacting proteins and be subjected to differential regulation. STAG2 may interact with oligodendrocyte-specific transcription factors and be preferentially recruited to myelination genes. It will be interesting to investigate the interactomes of STAG1 and STAG2 in oligodendrocytes using mass spectrometry.

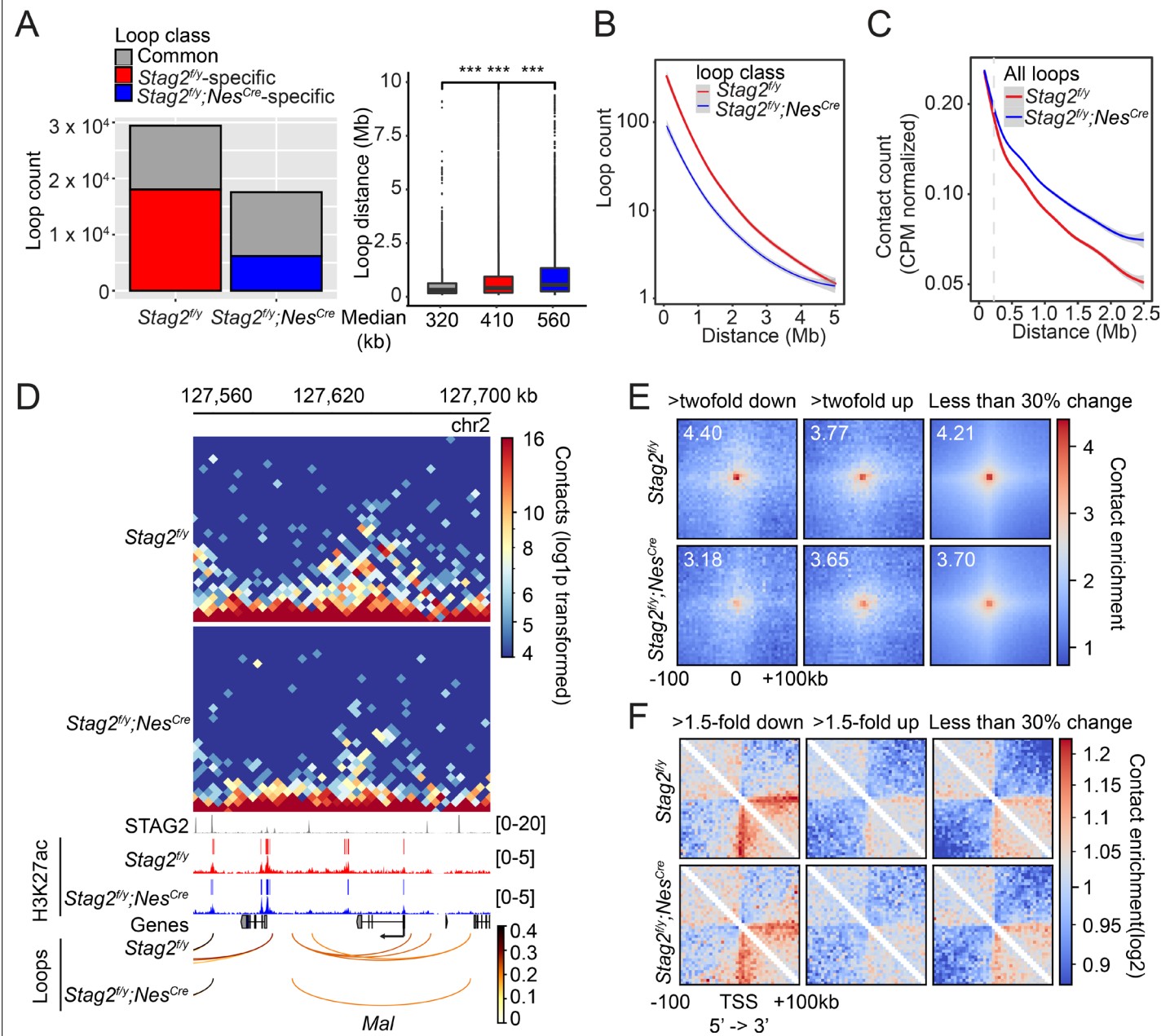

**Figure 7.** *Stag2* deletion impairs the formation of total and promoter-anchored loops in oligodendrocytes. (**A**) Loop counts (left panel) and length (right panel) in the indicated categories of *Stag2^{f/y}* and *Stag2^{f/y};Nes^{Cre}* oligodendrocytes (OLs). ***p < 0.001. (**B**) Loop counts plotted against loop length (from 0 to 5 Mb) of *Stag2^{f/y}* and *Stag2^{f/y};Nes^{Cre}* OLs. (**C**) Normalized contact counts for loops across different genomic distances in *Stag2^{f/y}* and *Stag2^{f/y};Nes^{Cre}* OLs. (**D**) Representative snapshots of contact maps at the *Mal* gene locus.hic files generated by HiC-Pro were converted to.cool format for plotting at 5 kb resolution. Tracks and narrow peaks from STAG2 and H3K27ac chromatin immunoprecipitation sequencing (ChIP-seq) as well as the loops are plotted below. Transcription direction is indicated by the black arrow. (**E**) Pile-up analysis of loop 'dots'-centered local maps for the promoter-anchored loops of genes in the indicated categories. The maps are balanced, normalized by distance, and plotted at 5 kb resolution. The numbers indicate the enrichment of the central pixel over the upper left and bottom right corners. (**F**) Pile-up analysis of the local contact maps centered around the transcription start site (TSS) of genes in the indicated categories. Transcription directions are indicated below. 1000 stable genes are chosen randomly and used for the analysis. The maps are balanced, normalized by distance, and plotted at 5 kb resolution. Diagonal pixels are omitted.

The online version of this article includes the following figure supplement(s) for figure 7:

**Figure supplement 1.** *Stag2* deletion reduces chromatin loops in oligodendrocytes.

**Figure supplement 2.** STAG2 controls local chromatin looping at differentially expressed genes.

**Figure supplement 3.** STAG2 regulates the formation of promoter-anchored loops in oligodendrocytes.

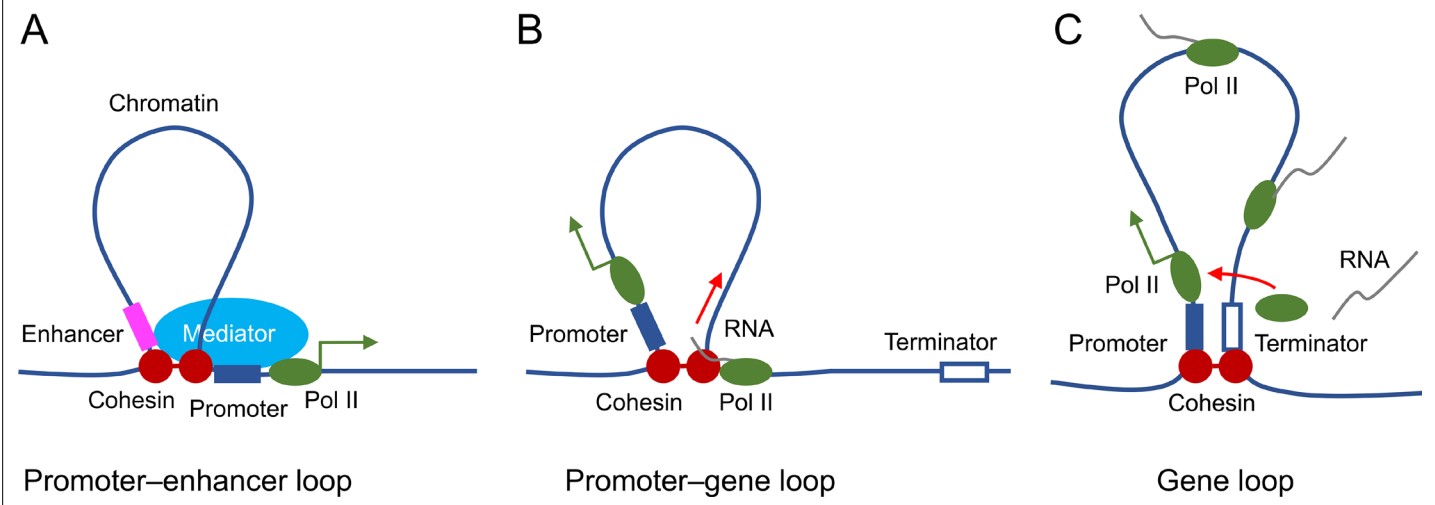

**Figure 8.** Proposed roles of STAG2-cohesin-mediated loop extrusion during transcription in oligodendrocytes. (**A**) STAG2-cohesin-mediated chromosome looping connects the enhancer and the promoter, thus facilitating interactions among oligodendrocyte-specific transcription factors, the mediator complex, and the general transcription machinery including RNA polymerase II. (**B**) STAG2-cohesin travels along the gene body via transcription-coupled loop extrusion to facilitate pre-mRNA processing. (**C**) STAG2-cohesin mediates the formation of gene loops that bring the terminator close to the promoter and facilitate Pol II recycling for multiple rounds of transcription.

## STAG2-mediated chromosome looping and transcription

The mechanisms by which STAG2-dependent chromosome looping facilitates transcription are unclear at present. We propose several models that are not mutually exclusive (*Figure 8*). First, by forming promoter–enhancer loops, STAG2-cohesin brings the mediator complex and other enhancer-binding factors to the spatial proximity of the general transcriptional machinery at the promoter, thereby enhancing RNA polymerase II recruitment and transcription initiation. The existence of P-stripes at STAG2-dependent genes in the Hi-C maps suggests that STAG2-mediated promoter–enhancer loops may involve enhancers located in the gene body. Second, loop extrusion by STAG2-cohesin may promote transcription elongation by regulating transcription-coupled pre-mRNA processing. For example, STAG2 has been shown to interact with RNA–DNA hybrid structures termed R-loops in vitro and in cells (*Pan et al., 2020*; *Porter et al., 2021*). R-loops formed between the nascent pre-mRNA and the DNA template impede transcription elongation and need to be suppressed (*Moore and Proudfoot, 2009*). When traveling with the transcription machinery on DNA, STAG2-cohesin might directly suppress R-loop formation or recruit other factors, such as the spliceosome, for cotranscriptional pre-mRNA processing and R-loop resolution. Third, STAG2-cohesin may establish promoter–terminator gene loops to recycle the RNA polymerase II that has finished one cycle of transcription back to the TSS for another round of transcription. Future experiments using high-resolution Hi-C methods in oligodendrocytes and ChIP-seq experiments with additional enhancer- and promoter-specific histone marks will allow us to better define the nature of STAG2-dependent promoter-anchored loops and stripes. It will also be interesting to examine whether *Stag2* deletion causes the accumulation of R-loops in downregulated genes and the incomplete splicing of their pre-mRNAs.

## Conclusion

We have discovered a requirement for the cohesin subunit STAG2 in the myelination of the CNS in mammals. Our findings implicate hypomyelination as a contributing factor to certain phenotypes of cohesinopathy, including growth retardation and neurological disorders. We provide evidence to suggest that STAG2 promotes the myelination transcriptional program in oligodendrocytes through the formation of promoter-anchored loops. Our study establishes oligodendrocytes as a physiologically relevant cell system for dissecting the cellular functions and regulatory mechanisms of cohesin-mediated chromosome folding and genome organization.

## Materials and methods

### Generation of mouse lines and mouse husbandry

All animals were handled in accordance with institutional guidelines of the Institutional Animal Care and Use Committee (IACUC; AAALAC unit number 000673) of University of Texas (UT) Southwestern Medical Center under the animal protocol number (APN) 102335. The *Stag2* locus was targeted by inserting one neo cassette and two loxP sites flanking exon 8 via homologous recombination in the mouse embryonic stem (ES) cells. G418-selected positive ES clones were screened for successful targeting by nested PCR tests on both 5′ and 3′ integration sites of loxP. Four confirmed ES clones were then microinjected into mouse blastocysts. The chimeras were bred to the R26FLP mouse line for the removal of the neo cassette. *Stag2^{f/+}* mice with the 129/B6 background were crossed with *Stag2^{f/y}* or WT C57BL/6J mice and maintained on this background. For the generation of the inducible system of *Stag2^{f/y};Rosa26^{CreErt2}* mice, *Stag2^{f/f}* mice were bred to the mouse strain that contains two alleles of the conditional Cre-ER^{T2} cassette (B6.129-Gt(ROSA)26Sortm1(cre/ERT2)Tyj/J, JAX stock #008463) (*Ventura et al., 2007*). For the generation of the nervous system-specific *Stag2^{f/y};Nes^{Cre}* mice, the *Stag2^{f/f}* mice were crossed with the transgenic mice carrying one allele of Cre recombinase driven by the rat nestin promoter and enhancer (Tg(Nes-cre)1Kln, JAX stock #003771) (*Giusti et al., 2014*; *Tronche et al., 1999*).

Whole-body knockout mice were generated by CRISPR–Cas9 gene editing technology. Briefly, a pair of guide RNAs (sgRNA; sequences listed in the Key Resource Table) targeting genomic sequence flanking exon 8 of the *Stag2* locus were tested for cutting efficiency in cell culture, transcribed in vitro, purified, checked for integrity, and microinjected into B6C3F1 mouse zygotes along with the *Cas9* mRNA (5-methylcytidine, pseudouridine, TriLink). 20 ng/µl of *Cas9* mRNA and 10 or 20 µg/µl each of sgRNA were used. The injected embryos were transferred to the surrogate mother on the same day. Mosaic $F_0$ founders carrying the *Stag2^{null}* allele were identified by PCR genotyping. The reduction of the STAG2 protein was confirmed by western blotting in multiple tissues. The $F_0$ founders were crossed with WT C57BL/6J mice to generate the *Stag2^{+/−}* $F_1$. The mutations in $F_1$ were identified by Sanger Sequencing. Two mouse lines carrying genomic deletions between the Cas9 cleavage sites were chosen for the generation of *Stag2^{null}* mouse embryos.

All mice were housed in the antigen-free barrier facility with 12 hr light/dark cycles (6 AM on and 6 PM off). Mice were fed a standard rodent chow (2016 Teklad Global 16% protein rodent diet, Harlan Laboratories).

### Immunoblotting

The C-terminal fragment of human STAG2 protein was expressed and purified from *Escherichia coli* and used as the antigen to generate rabbit polyclonal antibodies against STAG2 at YenZym. Other antibodies were purchased from the following commercial sources: anti-SMC1 (Bethyl Laboratories, A300-055A), anti-SMC3 (Bethyl Laboratories, A300-060A), anti-RAD21 (Bethyl Laboratories, A300-080A), anti-SA1 (Bethyl Laboratories, A302-579A), anti-SA2 (Bethyl Laboratories, A302-581A), anti-α-TUBULIN (Sigma-Aldrich, DM1A), anti-MBP (Abcam, ab7349), anti-PLP1 (Abcam, ab28486), and anti-H3K27ac (Abcam, ab4729).

For immunoblotting, brain hemispheres were homogenized in a Precellys tissue homogenizer (Bertin Instruments) with the lysis buffer [20 mM Tris–HCl (pH 7.7), 137 mM NaCl, 2 mM Ethylenediamine tetraacetic acid (EDTA), 10% (vol/vol) glycerol, 1% (vol/vol) Triton X-100, 0.5 mM dithiothreitol, 1 mM Phenylmethylsulfonyl fluoride (PMSF), 1 mM $Na_3VO_4$, 10 mM β-glycerophosphate, 5 mM NaF, and protease inhibitors (Roche)]. Homogenized brain tissues were lysed on ice for 1 hr. The lysate was then subjected to centrifugation at 20,817 × *g* at 4°C for 20 min and further cleared by filtering through a 0.45 µm filter. The cleared lysate was analyzed by sodium dodecyl sulfate–polyacrylamide gel electrophoresis and transferred to membranes, which was then incubated with the appropriate primary and secondary antibodies. The blots were imaged with the Odyssey Infrared Imaging System (LI-COR).

### Tissue histology and immunohistochemistry

Mouse brains were fixed in 10% neutral buffered formalin solution for 48 hr followed by paraffin embedding and coronal or sagittal sectioning at 5 µm. H&E staining and LFB staining were performed

by the Molecular Pathology Core at UT Southwestern Medical Center. Investigators were blinded to the genotype. Images were acquired with the DM2000 microscope (Leica) at ×1.25 resolution.

Immunohistochemistry was performed as previously described (*Choi et al., 2016*). Briefly, deparaffinized sections were fixed with 4% paraformaldehyde, subjected to antigen retrieval by boiling with 10 mM sodium citrate (pH 6.0), and then incubated with the indicated antibodies at 1:100 dilution. The slides were scanned with an Axioscan.Z1 microscope (Zeiss) at ×40 resolution at the Whole Brain Microscopy Facility at UT Southwestern Medical Center. Images were processed and quantified with ImageJ. For the myelinated fiber length measurement and coherency analysis, coronal sections of the brain cortex stained with the anti-MBP antibody were processed as previously described (*van Tilborg et al., 2017*). The myelinated axial thinning and fiber length measurement were performed by the plugin DiameterJ. The coherency analysis of myelinated axons was performed with the plugin OrientationJ.

## Isolation of primary oligodendrocytes

The immunomagnetic isolation of oligodendrocytes from *Stag2^{f/y}* and *Stag2^{f/y};Nes^{Cre}* P12-P14 pups was conducted using anti-O4 microbeads (Miltenyi Biotec) according to a published protocol (*Flores-Obando et al., 2018*). Briefly, brain cortices were dissected, pooled, minced into 1 mm³ cubes, and incubated with the Papain dissociation solution (neurobasal medium with 1% penicillin–streptomycin, 1% L-glutamine, 2% B27 supplement, 20–30 U/ml of Papain, and 2500 U DNase I) in a 37°C, 5% $CO_2$ incubator for more than 20 min. The enzymatic digestion was inactivated by the addition of 1 ml of fetal bovine serum (FBS). Gentle trituration by 10 ml, 5 ml and 1-ml pipettes was applied to break up cell clumps. Cells were collected by centrifugation (200 × *g*, 10 min), washed first with serum-containing medium (neurobasal medium with 1% penicillin–streptomycin, 1% L-glutamine, 2% B27 supplement, and 10% FBS), and then with the magnetic cell sorting (MCS) buffer (phosphate-buffered saline, pH 7.2, with 0.5% bovine serum albumin [BSA], 0.5 mM EDTA, 5 µg/ml insulin and 1 g/l glucose). The cell pellet was resuspended in the MCS buffer and incubated with anti-O4 microbeads at 10 µl/10⁷ cells at 4°C for 15 min followed by 1× wash with the MCS buffer. The O4⁺ immature oligodendrocytes were sorted through the magnetic LS columns according to the manufacturer's instruction. Freshly prepared oligodendrocytes were directly used or fixed for subsequent analysis.

## Metabolic cage analysis

Mice were singly housed in shoebox-sized cages with a 5-day acclimation period followed with a 4-day recording period. Recorded parameters were analyzed by the TSE system and normalized to body weight. The experiments were conducted by the core personnel under the core protocol at the Metabolic Phenotyping Core at UT Southwestern Medical Center. Investigators were blinded to the genotype.

## Growth hormone and IGF-1 detection

Blood samples were collected from facial bleeding without fasting. Plasma growth hormone levels were determined with the rat/mouse growth hormone ELISA kit (EMD Milipore, EZRMGH-45K). Plasma IGF-1 concentrations were measured using the mouse/rat IGF1 Quantikine ELISA kit (R&D Systems).

## Sterol and oxysterol composition analysis

Brain hemispheres were preweighed and snap-frozen for extraction and measurement by mass spectrometry. The sterol extraction and quantitative analysis were conducted at the Center of Human Nutrition at UT Southwestern Medical Center as described previously (*McDonald et al., 2012*).

## Electron microscopy

*Stag2^{f/y}* and *Stag2^{f/y};Nes^{Cre}* P18 pups were transcardially perfused with 4% paraformaldehyde, 1% glutaraldehyde in 0.1 M sodium cacodylate buffer (pH 7.4). Tissues were dissected and fixed with 2.5% (vol/vol) glutaraldehyde in 0.1 M sodium cacodylate buffer (pH 7.4) for at least 2 hr. After three rinses with the 0.1 M sodium cacodylate buffer, optic nerve samples were embedded in 3% agarose and sliced into small blocks. All samples were again rinsed with the 0.1 M sodium cacodylate buffer three times and postfixed with 1% osmium tetroxide and 0.8% potassium ferricyanide in the 0.1 M

sodium cacodylate buffer for 3 hr at room temperature. Blocks were rinsed with water and *en bloc* stained with 4% uranyl acetate in 50% ethanol for 2 hr. Samples were dehydrated with increasing concentrations of ethanol, transitioned into propylene oxide, infiltrated with Embed-812 resin, and polymerized in a 60°C oven overnight. Blocks were sectioned with a diamond knife (Diatome) on a Leica Ultracut 7 ultramicrotome (Leica Microsystems) and collected onto copper grids, poststained with 2% aqueous uranyl acetate and lead citrate. Images were acquired on a Tecnai G2 Spirit transmission electron microscope (Thermo Fisher) equipped with a LaB6 source using a voltage of 120 kV. Tissue processing, sectioning, and staining were completed by the Electron Microscopy Core at UT Southwestern Medical Center.

## RNA-seq library preparation and sequencing

Total RNA was extracted from brain hemispheres or isolated oligodendrocytes with Trizol. RNA integrity was determined by the Agilent BioAnalyzer 2100. TruSeq Stranded mRNA library prep kit (Illumina) was used to generate the mRNA libraries. The libraries were analyzed by the Bioanalyzer and multiplexed and sequenced using the NextSeq 500 high output kit (400 M reads) for the brain libraries or NextSeq 500 mid output kit (130 M reads) for the isolated oligodendrocytes libraries at the Next Generation Sequencing Core at UT Southwestern Medical Center.

## Differential expression and pathway analysis

Raw data from the sequencer were demultiplexed and converted to fastq files using bcl2fastq (v2.17, Illumina). The fastq files were checked for quality using fastqc (v0.11.2) *Andrews, 2010* and fastq_screen (v0.4.4) (*Wingett, 2011*). Fastq files were mapped to the mm10 mouse reference genome (from iGenomes) using STAR (*Dobin et al., 2013*). Read counts were then generated using featureCounts (*Liao et al., 2014*). TMM normalization and differential expression analysis were performed using edgeR (*Robinson et al., 2010*). Pathway analysis was performed with the IPA software. Genes with more than 1.5-fold change and false-discovery rate FDR <0.01 were included in the brain RNA-seq pathway analysis. Genes with more than twofold change and FDR <0.05 were used for the pathway analysis of the RNA-seq data from oligodendrocytes.

## RT-qPCR analysis

Single-stranded cDNAs were converted from 2 µg of total RNA extracted from mouse brains with the high-capacity cDNA reverse transcription kit (Applied Biosystems). Quantitative PCR was conducted to determine transcript levels using gene-specific TaqMan probes (Applied Biosystems).

## Single-cell RNA-seq

Single-cell suspension was prepared from forebrains of P13 *Stag2^f/y^* or *Stag2^f/y^;Nes^Cre^* pups using the Papain Dissociation System (Worthington Biochemical, LK003150) according to the manufacturer's instructions. Biological duplicates were made for each genotype. Single-cell RNA-seq libraries were generated with the Chromium Single Cell 3' GEM, Library & Gel Bead Kit v3 (10× Genomics) according to the manufacturer's guidelines. Cell density and viability were checked by the TC-20 Cell Counter (Bio-Rad). Cells were then loaded onto Chip B in the Chromium Controller (10× Genomics). 10,000 cells were targeted for each sample. The libraries were analyzed by the Bioanalyzer (Agilent) and pair-end sequenced in two flowcells of the NextSeq 500 High Output (400 M) run. The sequencing was performed at the Next Generation Sequencing Core at UT Southwestern Medical Center.

Data demultiplexing and alignment were performed using the Cell Ranger pipeline (https://support.10xgenomics.com/single-cell-gene-expression/software/pipelines/latest/using/ mkfastq) (10× Genomics). The raw features, barcodes, and matrixes were used as input for further analysis using the R package Seurat3 (*Butler et al., 2018*; *Stuart et al., 2019*) (https://satijalab.org/seurat/). Cells were filtered by the following criteria: nFeature_RNA (200–9500) and percent.mt <10. After filtering, a total of 5834 cells in *Stag2^f/y^#1*, 4699 cells in *Stag2^f/y^#2*, 9050 cells in *Stag2^f/y^;Nes^Cre^#1*, and 3073 cells in *Stag2^f/y^;Nes^Cre^#2* were used for downstream analysis. 2000 variable features were found from each normalized dataset. All datasets were then integrated using identified anchors (dims = 1:30). Standard scaling and principal component analysis, clustering (resolution = 0.5), and tSNE reduction (dims = 1:30) were performed on the integrated dataset. Cluster biomarkers were identified, and top features were examined. Clusters were then manually assigned to distinct cell-type identities with knowledge

from previous studies (*Cahoy et al., 2008*; *Dulken et al., 2019*; *Marques et al., 2018*; *Marques et al., 2016*; *Marton et al., 2019*; *Saunders et al., 2018*; *Zeisel et al., 2018*; *Zywitza et al., 2018*) (http://www.brainrnaseq.org/) (http://dropviz.org/). Clusters with the same cell-type identities were merged. Five clusters of oligodendrocyte lineage [cycling oligodendrocyte progenitors (OPCcycs), oligodendrocyte progenitors (OPCs), newly formed oligodendrocytes (NFOLs), myelin-forming oligo-dendrocytes (mFOLs), and fully matured oligodendrocytes (MFOLs)] were identified and selected for indicated gene expression comparison and plotting using Vlnplot or FeaturePlot functions. The trajectory analysis was performed using Monocle3 (*Cao et al., 2019*) in the oligodendrocyte cell population. Gene density plot over pseudotime was generated as previously described (*Luecken and Theis, 2019*).

## ChIP-seq

Chromatin immunoprecipitation (ChIP) was performed as previously described (*Liu et al., 2017*). Briefly, isolated oligodendrocytes were fixed with 1% formaldehyde and fragmented with a soni-cator (Branson 450). The fragmented chromatin was incubated with antibodies overnight at 4°C. Dynabeads Protein A (Thermo Fisher Scientific) was used for the immunoprecipitation. Libraries were generated by the Next Gen DNA Library Kit (Active Motif) with the Next Gen Indexing Kit (Active Motif) for STAG2 ChIP-seq or the KAPA HyperPrep Kits (KAPA Systems) for histone ChIP-seq. The libraries were analyzed by the Bioanalyzer and pool sequenced with the NextSeq 500 mid output (130 M) kit. After mapping reads to the mouse genome (mm10) by bowtie2 (v2.2.3) (*Langmead and Salzberg, 2012*) with the parameter '–sensitive', we performed filtering by removing alignments with mapping quality less than 10 and then removing duplicate reads identified by Picard MarkDuplicates (v1.127). For STAG2 ChIP-seq, Picard MarkDuplicate was used to remove duplicates together with options to use molecular identifiers (MIDs) information in the reads. Enriched regions (peaks) were identified using MACS2 (v2.0.10) (*Zhang et al., 2008*), with a *q*-value cutoff of 0.05 for peaks. Peak regions were annotated by HOMER (*Ross-Innes et al., 2012*).

## Hi-C library generation, sequencing, and analysis

Hi-C was performed at the Genome Technology Center at NYU Langone Health from 3.5 to 4.0 µg of DNA isolated from cells cross-linked with 2% formaldehyde at room temperature for 10 min. Experiments were performed in duplicates following the instructions from the Arima Hi-C kit (Arima Genomics, San Diego, CA). Subsequently, Illumina-compatible sequencing libraries were prepared by using a modified version of the KAPA HyperPrep library kit (KAPA BioSystems, Willmington, MA). Quality check steps were performed to assess the fraction of proximally ligated DNA labeled with biotin, and the optimal number of PCR reactions needed to make libraries. The libraries were loaded into an Illumina flowcell (Illumina, San Diego, CA) on a NovaSeq 6000 instrument for paired-end 50 reads.

Hi-C analysis was performed using the HiC-Bench pipeline (*Lazaris et al., 2017*; *Tsirigos et al., 2012*) (https://github.com/NYU-BFX/hic-bench) and HiC-Pro v3.1.0 (*Servant et al., 2015*). The read pairs were aligned and filtered with the following parameters: Genome-build=mm10; –very-sensitive-local –local; mapq = 20; –min-dist 25000 –max-offset 500. The Juicer 'pre' tool (*Durand et al., 2016*) (RRID: SCR_017226, v1.11.09; https://github.com/aidenlab/juicer) was used to generate the.hic file with default parameters. Sample duplicates were combined. The compartment analysis was done using the HOMER tool (*Heinz et al., 2010*) (http://homer.ucsd.edu/homer/index. html) with 100 kb bins. H3K27ac ChIP-seq data were used to assign A/B compartments. Eigen-vector-1 bins were considered shifted (AB and BA) when the bin sign changed and the delta value was greater than 1.5. TADs and boundaries were identified at 40 kb resolution with the HiCRatio method with the follow parameters: –min-lambda=0.0 –max-lambda=1.0 –n-lambda=6 –gamma = 0 –distance = 500 kb –fdr = 0.1. TADs were also identified using the Juicer tools (v1.22.01) arrowhead at 10 and 25 kb resolution. Aggregate TAD analysis was performed on TAD boundaries by coolpup. py (*Flyamer et al., 2020*) or GENOVA (*van der Weide et al., 2021*). The.hic files were converted to.cool format for visualization and plotting with pyGenomeTracks (*Lopez-Delisle et al., 2021*) at 5 kb resolution.

## Loop analysis and RNA-seq integration

The loops were classified into group-specific loops and common loops by using the significance cutoffs provided by Fit-HiC (*Ay et al., 2014*). A $q$-value cutoff of 0.01 was used to identify significant loops in both groups. A loop is considered 'group-specific' if it is only present in one group with a $q$ value <0.01 and not present in the other group with cutoff of $q$ val <0.1. Loop anchors were annotated with the gene promoter information (promoter defined as ±2 kb from the TSS). The genes were classified into 'down' and 'up' regulated genes using an FDR cutoff of 0.05, logFC cutoff of ±0.58 and logCPM >0. 'stable' or less changed genes are defined as logFC <0.38, and logCPM >0. Random 1000 genes were chosen for analysis and plotting. The active genes (logCPM >0) were also grouped in 'high', 'mid', and 'low' expression groups by separating the genes in three quantiles according to the logCPM values. For the loop enrichment scores, normalized contact scores were computed using Fit-HiC at 10 kb resolution and bias corrected. Pile-up analysis was performed with coolpup.py (*Flyamer et al., 2020*) with the KR method to balance the weight and random shift controls for distance normalization at 5 kb or using GENOVA.

## Acknowledgements

We thank Sung Jun Bae for taking the mouse photos and John Shelton for help with histology and in situ hybridization. We are grateful to Jeffrey McDonald for the sterol composition analysis, Richard Lu and Lu Sun for providing reagents and advice for the isolation of oligodendrocytes, and Applied Bioinformatics Laboratories at NYU Langone Health for the Hi-C analysis. We also thank the Yu lab members for helpful discussions and for reading the manuscript critically. This study was supported by the National Natural Science Foundation of China (Project 32130053), the U.S. National Institutes of Health (1R01GM124096), the Cancer Prevention and Research Institute of Texas (CPRIT) (RP160667-P2), and the Welch foundation (I-1441).

## Additional information

### Funding

| Funder | Grant reference number | Author |
| --- | --- | --- |
| National Natural Science Foundation of China | Project 32130053 | Hongtao Yu |
| National Institutes of Health | 1R01GM124096 | Hongtao Yu |
| Cancer Prevention and Research Institute of Texas | RP160667-P2 | Hongtao Yu |
| Welch Foundation | I-1441 | Hongtao Yu |

The funders had no role in study design, data collection, and interpretation, or the decision to submit the work for publication.

### Author contributions

Ningyan Cheng, Conceptualization, Data curation, Formal analysis, Supervision, Funding acquisition, Investigation, Methodology, Writing - original draft, Project administration, Writing - review and editing; Guanchen Li, Mohammed Kanchwala, Formal analysis, Visualization, Methodology; Bret M Evers, Formal analysis, Visualization, Methodology, Writing - review and editing; Chao Xing, Formal analysis, Supervision, Funding acquisition, Project administration; Hongtao Yu, Conceptualization, Software, Supervision, Funding acquisition, Methodology, Project administration, Writing - review and editing

### Author ORCIDs

Ningyan Cheng http://orcid.org/0000-0001-8764-552X
Bret M Evers http://orcid.org/0000-0001-5686-0315
Chao Xing http://orcid.org/0000-0002-1838-0502

Hongtao Yu http://orcid.org/0000-0002-8861-049X

### Ethics

All animals were handled in accordance with institutional guidelines of the Institutional Animal Care and Use Committee (IACUC; AAALAC unit number 000673) of University of Texas (UT) Southwestern Medical Center under the animal protocol number (APN) 102335.

### Decision letter and Author response

Decision letter https://doi.org/10.7554/eLife.77848.sa1
Author response https://doi.org/10.7554/eLife.77848.sa2

---

## Additional files

### Supplementary files

• Supplementary file 1. List of enriched pathways of differentially expressed genes between wild-type (WT) and *Stag2* KO mouse brains as revealed by ingenuity pathway analysis (IPA).

• Supplementary file 2. List of differentially expressed genes between wild-type (WT) and *Stag2* KO oligodendrocytes, with the status of STAG2 binding at their promoters and the numbers of promoter-anchored loops indicated.

• Transparent reporting form

### Data availability

The RNA-seq, scRNA-seq, ChIP-seq, and Hi-C datasets generated and analyzed during the current study are available in the GEO repository, with the accession number GSE186894.

The following dataset was generated:

| Author(s) | Year | Dataset title | Dataset URL | Database and Identifier |
|---|---|---|---|---|
| Cheng N, Kanchwala M, Evers BM, Xing C, Yu H | 2021 | STAG2 promotes the myelination transcriptional program in oligodendrocytes | https://www.ncbi.nlm.nih.gov/geo/query/acc.cgi?acc=GSE186894 | NCBI Gene Expression Omnibus, GSE186894 |

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

# Appendix 1

### Appendix 1—key resources table

| Reagent type (species) or resource | Designation | Source or reference | Identifiers | Additional information |
|---|---|---|---|---|
| Strain, strain background (*Mus musculus*, female) | *Stag2⁺/⁻* | This paper | | Exon8 of *Stag2* was targeted by CRISPR–Cas9 (see Materials and methods) |
| Strain, strain background (*Mus. musculus*, both sex) | *Stag2^f/y^; Stag2^f/f^* | This paper | | Exon8 of *Stag2* genomic locus was flanked by loxP sites (see Materials and methods) |
| Strain, strain background (*Mus. musculus*, both sex) | C57BL/6J | The Jackson Laboratory | 000664; RRID:IMSR_JAX:000664 | |
| Strain, strain background (*Mus. musculus*, both sex) | B6.129-Gt(ROSA)26 Sortm1(cre/ERT2)Tyj/J | The Jackson Laboratory | 008463; RRID:IMSR_JAX:008463 | |
| Strain, strain background (*Mus. musculus*, male) | B6.Cg-Tg(Nes-cre)1Kln/J | The Jackson Laboratory | 003771; RRID:IMSR_JAX:003771 | |
| Antibody | anti-STAG2 (Rabbit polyclonal) | This paper | | The C-terminus recombinant protein of STAG2 (*Homo sapiens*) was used to generate the antibody; WB (1:1000) |
| Antibody | anti-α-TUBULIN (Mouse monoclonal) | Sigma-Aldrich | T9026; RRID:AB_477593 | WB (1:1000) |
| Antibody | anti-SA1 (Rabbit polyclonal) | Bethyl Laboratories | A302-579A; RRID:AB_2034857 | WB (1:1000) |
| Antibody | anti-SMC1 (Rabbit polyclonal) | Bethyl Laboratories | A300-055A RRID:AB_2192467 | WB (1:1000) |
| Antibody | anti-SMC3 (Rabbit polyclonal) | Bethyl Laboratories | A300-060A; RRID:AB_67579 | WB (1:1000) |
| Antibody | anti-RAD21 (Rabbit polyclonal) | Bethyl Laboratories | A300-080a; RRID:AB_2176615 | WB (1:1000) |
| Antibody | anti-MBP (Rat monoclonal) | Abcam | ab7349; RRID:AB_305869 | IHC (1:100) |
| Antibody | anti-PLP1 (Rabbit polyclonal) | Abcam | ab28486; RRID:AB_776593 | IHC (1:100) |
| Antibody | anti-GFAP (Rabbit polyclonal) | Abcam | ab7260; RRID:AB_305808 | IHC (1:100) |
| Antibody | anti-MAP2 (Rabbit polyclonal) | Abcam | ab32454; RRID:AB_776174 | IHC (1:50) |
| Antibody | anti-H3K27ac (Rabbit polyclonal) | Abcam | ab4729; RRID:AB_2118291 | ChIP (5 µl per test) |
| Antibody | anti-O4 Microbeads (Mouse monoclonal) | Miltenyi Biotec | 130-094-543; RRID:AB_2847907 | MACS (10 µl per 10⁷ cells) |
| Antibody | anti-rabbit IgG (H+L), DyLight 800 Conjugate (Goat polyclonal) | Cell Signaling Technology | 5151 S; RRID:AB_10697505 | WB (1:5000) |

*Appendix 1 Continued on next page*

*Appendix 1 Continued*

| Reagent type (species) or resource | Designation | Source or reference | Identifiers | Additional information |
|---|---|---|---|---|
| Antibody | anti-mouse IgG (H+L), DyLight 680 Conjugate (Goat polyclonal) | Cell Signaling Technology | 5470 S; AB_10696895 | WB (1:5000) |
| Antibody | anti-rat IgG (H+L), Alexa Fluor 568 (Goat polyclonal) | Thermo Fisher Scientific | A-11077; RRID:AB_2534121 | IHC (1:500) |
| Antibody | anti-rabbit IgG (H+L), Alexa Fluor 488 (Goat polyclonal) | Thermo Fisher Scientific | A-11008; RRID:AB_143165 | IHC(1:500) |
| Sequence-based reagent | sgRNA#1 target on *Stag2* | This paper | CRISPR single-guide RNA target sequence | Target sequence: TAGCCAACCTCTTTCT CTATTGG |
| Sequence-based reagent | sgRNA#2 target on *Stag2* | This paper | CRISPR single-guide RNA target sequence | Target sequence: CAGACAGTATACTGTAATGGAGG |
| Sequence-based reagent | TaqMan probes: *Stag2* | Thermo Fisher Scientific | Mm01311611_m1 | |
| Sequence-based reagent | TaqMan probes: *Klk6* | Thermo Fisher Scientific | Mm00478322_m1 | |
| Sequence-based reagent | TaqMan probes: *Ninj2* | Thermo Fisher Scientific | Mm00450216_m1 | |
| Sequence-based reagent | TaqMan probes: *Cpm* | Thermo Fisher Scientific | Mm01250802_m1 | |
| Sequence-based reagent | TaqMan probes: *Fa2h* | Thermo Fisher Scientific | Mm00626259_m1 | |
| Sequence-based reagent | TaqMan probes: *Gapdh* | Thermo Fisher Scientific | Mm99999915_g1 | |
| Sequence-based reagent | Stag2 gt 5 F | This paper | Genotype sequence primers | GGTATTTACTTGATAGCCAACC |
| Sequence-based reagent | Stag2 gt 5 R | This paper | Genotype sequence primers | CTCATCTTGATTTTCCTGAAGC |
| Sequence-based reagent | Stag2 gt 3 F | This paper | Genotype sequence primers | GGTTGAGACAGACAGTATAC |
| Sequence-based reagent | Stag2 gt 3 R | This paper | Genotype sequence primers | AGGCTGGACTATGACAACTC |
| Sequence-based reagent | ISH Probe Stag2 P1 F | This paper | Riboprobe synthesis primers | TACGGTACCGACCTTTCAGATGTC ACTCCG |
| Sequence-based reagent | ISH Probe Stag2 P1 R | This paper | Riboprobe synthesis primers | GAAGGATCCGCATCGGATAGACAC TCATGA |
| Sequence-based reagent | ISH Probe Stag2 P2 F | This paper | Riboprobe synthesis primers | TACGGATCCGACCTTTCAGATGTC ACTCCG |
| Sequence-based reagent | ISH Probe Stag2 P2 R | This paper | Riboprobe synthesis primers | GAAGGTACCGCATCGGATAGACAC TCATGA |
| Sequence-based reagent | ISH Probe Stag1 P1 F | This paper | Riboprobe synthesis primers | TTAGGTACCTTACAATGCCTGGTCCTCAGT |
| Sequence-based reagent | ISH Probe Stag1 P1 R | This paper | Riboprobe synthesis primers | GAAGGATCCCTTTCATTGGCTCTCTTCCC |
| Sequence-based reagent | ISH Probe Stag1 P2 F | This paper | Riboprobe synthesis primers | TTAGGATCCTTACAATGCCTGGTCCTCAGT |
| Sequence-based reagent | ISH Probe Stag1 P2 R | This paper | Riboprobe synthesis primers | GAAGGTACCCTTTCATTGGCTCTCTTCCC |
| Commercial assay or kit | Arima-HiC Kit | Arima Genomics | 510008 | |

*Appendix 1 Continued on next page*

*Appendix 1 Continued*

| Reagent type (species) or resource | Designation | Source or reference | Identifiers | Additional information |
|---|---|---|---|---|
| Chemical compound, drug | Tamoxifen | Sigma-Aldrich | T5648 | |
| Chemical compound, drug | 4-Hydroxytamoxifen | Sigma-Aldrich | H7904 | |
| Software, algorithm | GraphPad Prism | GraphPad Software | RRID:SCR_002798; https://www.graphpad.com/scientific-software/prism/ | |
| Software, algorithm | ImageJ (Fiji) | ImageJ | RRID:SCR_002285; https://imagej.net/software/fiji/ | |
| Software, algorithm | RStudio | The R Foundation | RRID:SCR_000432; https://www.rstudio.com/ | |
| Software, algorithm | Bcl2fastq | Illumina | RRID:SCR_015058 | v2.17 |
| Software, algorithm | Fastqc | *Andrews, 2010*; PMID:24501021 | RRID:SCR_014583 | v0.11.2 |
| Software, algorithm | Fastq_screen | *Wingett, 2011* | RRID:SCR_000141; https://www.bioinformatics.babraham.ac.uk/projects/fastqc/ | v0.4.4 |
| Software, algorithm | STAR | *Dobin et al., 2013*; PMID:23104886 | RRID:SCR_004463; https://github.com/alexdobin/STAR | v2.5.3a |
| Software, algorithm | FeatureCounts | *Liao et al., 2014*; PMID:24227677 | RRID:SCR_012919; https://bioconductor.org/packages/release/bioc/html/Rsubread.html | |
| Software, algorithm | edgeR | *Robinson et al., 2010*; PMID:19910308 | RRID:SCR_012802; https://bioconductor.org/packages/release/bioc/html/edgeR.html | |
| Software, algorithm | Ingenuity pathway analysis | QIAGEN, *Krämer et al., 2014*; PMID:24336805 | RRID:SCR_008653; https://www.qiagenbioinformatics.com/products/ingenuity-pathway-analysis | |
| Software, algorithm | MACS2 | *Zhang et al., 2008*; PMID:18798982 | RRID:SCR_013291 | v2.0.10 |
| Software, algorithm | Bowtie2 | *Langmead and Salzberg, 2012*; PMID:22388286 | RRID:SCR_016368 | v2.2.3 |
| Software, algorithm | Picard MarkDuplicates | Broad Institute, GitHub Repository | RRID:SCR_006525; http://broadinstitute.github.io/picard/ | v1.127 |
| Software, algorithm | HOMER | *Heinz et al., 2010*, *Ross-Innes et al., 2012*; PMID:20513432 | RRID:SCR_010881; http://homer.ucsd.edu/homer/ | |

*Appendix 1 Continued on next page*

*Appendix 1 Continued*

| Reagent type (species) or resource | Designation | Source or reference | Identifiers | Additional information |
|---|---|---|---|---|
| Software, algorithm | Deeptools | *Ramírez et al., 2016*; PMID:27079975 | RRID:SCR_016366; https://deeptools.readthedocs.io/en/develop/ | |
| Software, algorithm | Galaxy | *Afgan et al., 2018*; PMID:29790989 | RRID:SCR_006281; https://usegalaxy.org | |
| Software, algorithm | Cell Ranger | 10× Genomics | RRID:SCR_017344; https://support.10xgenomics.com/single-cell-gene-expression/software/pipelines/latest/using/mkfastq | |
| Software, algorithm | Seurat | New York Genome Center; *Stuart et al., 2019*; PMID:31178118 | RRID:SCR_016341; https://satijalab.org/seurat | Satija Lab |
| Software, algorithm | Monocle3 | UW Genome Sciences; *Cao et al., 2019*; PMID:30787437 | RRID:SCR_018685; https://cole-trapnell-lab.github.io/monocle3/ | Cole Trapnell's Lab, v3.0 |
| Software, algorithm | HiC-Bench pipeline | *Lazaris et al., 2017, Tsirigos et al., 2012*; PMID:22113082 | https://github.com/NYU-BFX/hic-bench | v0.1 |
| Software, algorithm | Juicer 'pre' tool | *Durand et al., 2016*; PMID:27467249 | RRID:SCR_017226; https://github.com/aidenlab/juicer | Aiden Lab, v1.11.09 |
| Software, algorithm | Juicebox | Aiden Lab, BCM | RRID:SCR_021172; https://github.com/aidenlab/Juicebox | v1.5.1 |
| Software, algorithm | Hic2cool | *Abdennur and Mirny, 2020*; PMID:31290943 | https://github.com/4dn-dcic/hic2cool | v0.8.3 |
| Software, algorithm | pyGenomeTracks | *Lopez-Delisle et al., 2021*; PMID:32745185 | https://github.com/deeptools/pyGenomeTracks | v3.7 |
| Software, algorithm | Fit-HiC | *Ay et al., 2014*; PMID:24501021 | https://github.com/ay-lab/fithic | v2.0.7 |
| Software, algorithm | Coolpup.py | *Flyamer et al., 2020*; PMID:32003791 | https://github.com/open2c/coolpuppy | v0.9.5 |
| Software, algorithm | clusterProfiler | Bioinformatics Group, Southern Medical University; *Wu et al., 2021*; PMID:34557778 | RRID:SCR_016884; https://github.com/YuLab-SMU/clusterProfiler | v4.4.1 |
| Software, algorithm | HiC-Pro | *Servant et al., 2015*; PMID:26619908 | RRID:SCR_017643 | v3.1.0 |
| Software, algorithm | HiCRep | *Yang et al., 2017*; PMID:28855260 | https://github.com/TaoYang-dev/hicrep | v1.11.0 |
| Software, algorithm | GENOVA | *van der Weide et al., 2021*; PMID:34046591 | https://github.com/robinweide/GENOVA | v1.0.0.9 |

