## [Editor Report]

This manuscript will be of interest to scientists working on genome organisation and transcriptional control of myelination during mammalian brain development. The authors combine diverse and complementary experimental approaches to generate insights into how DNA looping contributes to transcriptional regulation in functionally specialised cell types. The experiments have been rigorously performed and the main conclusions are justified.

---

## [Decision Letter]

**Decision letter after peer review:**

Thank you for submitting your article "STAG2 promotes the myelination transcriptional program in oligodendrocytes" for consideration by *eLife*. Your article has been reviewed by 3 peer reviewers, including Adèle L Marston as the Reviewing Editor and Reviewer #1, and the evaluation has been overseen by Jessica Tyler as the Senior Editor. The following individuals involved in the review of your submission have agreed to reveal their identity: Andrew J Wood (Reviewer #2); Simone Di Giovanni (Reviewer #3).

Essential revisions:

1. Although the NexCre system is widely used, there is no clear consensus in the literature where the Cre recombinase is expressed. While Giusti et al. 2014 (J. of psychiatric research) reported neuronal and astroglial expression, Goebbels et al. 2006 (Genesis) reported neuronal only expression. A more detailed characterization of the cell type specificity and timing of Stag2 expression and ablation in the NPCs lineage would be useful. This is important to understand whether OLs are particularly sensitive to Stag2 deletion or whether Stag2 has not been deleted in neurons and astrocytes. From Figure 4A and FigS6B, neurons and astrocytes are hardly retrieved from WT brains, so it is difficult to compare Stag2 expression level and ablation in these cells. While it is clear that OLs are affected by Stag2 KO, it is not certain that these are the only and main cell type affected. Along the same lines, since it is difficult to quantify cell composition from the RNAseq, due to different cell survival and purification, a better characterization of cell type numbers on histological section is needed to clarify if Stag2Cre mice have indeed a generally normal neuronal cell differentiation.

2. More details are required for the 3D chromatin structure analysis. Although the data are mainly in line with the literature, opposite to other work (Wutz et al., EMBOJ 2017; Pekowska et al., Nature 2017; Casa et al., Genome Res 2020), the authors did not find any TAD alterations. This might be related to the resolution used; the methods or legends do not state which resolution has been used. Please clarify and include this information in the figure legend, and make sure that at least 25 kb resolution is used for TAD analysis. You should report a summary of the sequencing data including the total reads obtained and, importantly the number of valid cis-pairs in each of their libraries etc.

3. Also for the Hi-C data: from Figure 5F and 5E, the correlation between the 2 biological replicates seems to be not very high; please comment. Since the effect is only on the loops and is quite modest, it is important that you demonstrate that it is reproducible and report whether this effect was seen in independent datasets, and whether the same loops were affected in each case.

4. Regarding the single-cell and purified OL RNAseq, only two biological replicates have been used, but three would be recommended.

5. Please provide a better discussion of what distinguishes genes that are or are not regulated by STAG2. On page 15, you speculate that STAG2 might interact with oligodendrocyte-specific transcription factors and be preferentially recruited to myelination genes. Preferential recruitment to myelination genes should be tested using the existing ChIP-seq data represented in Figure S8.

6. Similarly, all three of the models shown in Figure 7 appear to indicate that STAG2 ordinarily regulates transcription through the formation of promoter-anchored loops, via a mechanism that involves direct binding to the relevant promoter in question. STAG2 binding sites should therefore be enriched at promoters where loops are lost in the STAG2 mutant condition. You should determine whether this is the case and discuss the implications for their models if it is not.

7. Similarly, you hypothesise three possible ways for Stag2 mediated gene expression regulation. One is by assisting enhancer-promoter loops. The authors have generated H3K27ac Chip from purified OLs, and this could be used to map E-P loops and test their hypothesis.

8., Figure 6D – A gene with a better-defined role in myelination than Pls1 would be preferable to use as an exemplar locus here.

8. Please provide the list of DE genes, STAG2 genomic occupancy, or promoters anchored loops as a supplementary file. This will help the readability of the paper and will allow full disclosure of the dataset to the community. Furthermore, please provide the reviewer passkey for the GEO link.

10. The authors are recommended to perform GO and pathway analysis using some other tools in addition to IPA, like GSEA. Also, since the pathways shown in the chart in Fig2Da and 4G are the top enriched pathways, it would be useful if the authors could provide a list of all the enriched pathways. Furthermore, please clarify what you used as background for the enrichment analysis.

11. From the purified OL RNAseq, the authors found 271 down and 292 upregulated genes. Please clarify what these genes are. In Fig4G, you showed a pathway analysis of all the DE, and it is not clear which gene set has been used for FigS7E. It would be useful to characterize the UP and DOWN genes separately.

12. In the single-cell RNAseq, you have used nFeature_RNA(200-9500). This seems to be a very high threshold, with the risk to include cell doublets. Please disclose whether you can reproduce the same analysis using a lower threshold.

13. It would be useful to test using the Stag2 ChIP data in isolated OLs, whether loop loss is observed preferentially on Stag2 occupied sites and how STAg2 occupancy correlates with loop score of up/down/no DE genes.

14. A more detailed Methods section is required: for example, how many replicates for the sequencing, sequencing depth, the sequence of the probes, PCR primers, antibodies (codes and quantity); more details for the OL isolation procedure. Methods for Cas9 mediated KO are missing.

15. Similarly, Figure Legends need more details for clarity

*Reviewer #2 (Recommendations for the authors):*

(1) Page 8 – the statement " a mild reduction in the number of MOLs" should be changed to "a mild reduction in the proportion of MOLs"

(2) Page 12 contains the statement "Highly expressed genes might be more reliant on these loops for transcription and are preferentially downregulated by Stag2 loss". Please provide a reference to the data which support the second part of this statement.

---

## [Author Response]

Essential revisions:1. Although the NexCre system is widely used, there is no clear consensus in the literature where the Cre recombinase is expressed. While Giusti et al. 2014 (J. of psychiatric research) reported neuronal and astroglial expression, Goebbels et al. 2006 (Genesis) reported neuronal only expression. A more detailed characterization of the cell type specificity and timing of Stag2 expression and ablation in the NPCs lineage would be useful. This is important to understand whether OLs are particularly sensitive to Stag2 deletion or whether Stag2 has not been deleted in neurons and astrocytes. From Figure 4A and FigS6B, neurons and astrocytes are hardly retrieved from WT brains, so it is difficult to compare Stag2 expression level and ablation in these cells. While it is clear that OLs are affected by Stag2 KO, it is not certain that these are the only and main cell type affected. Along the same lines, since it is difficult to quantify cell composition from the RNAseq, due to different cell survival and purification, a better characterization of cell type numbers on histological section is needed to clarify if Stag2Cre mice have indeed a generally normal neuronal cell differentiation.

In the initial submission, we have shown in the original Figure S6B,C that *Stag2* expression was robust in most cell types in the brain and its level was greatly reduced in neuronal lineages in *Nes^Cre^* mice. Based on single-cell RNA-seq, the levels of *Stag2* transcripts were reduced in “aNSCS/NPCs”, “Astrocytes/qNSCs”, “Astrocytes”, “Neuroblasts”, and OLs populations, but not in non-neural cell clusters, such as “Endothelials” and “Microglia/Macrophages”. As suggested by the reviewers, we examined the expression changes of *Stag2* and other cohesin genes in OLs, astrocytes, and neuronal lineages (Figure 4—figure supplement 1). Based on this analysis, it is clear that *Stag2* ablation occurred early in the NPC stage and was maintained in all differentiated cell lineages. We agree that the numbers of neurons and astrocytes in the wild-type sample were low. As the genetic ablation of *Stag2* already occurred at the NPC stage, it is very likely that *Stag2* ablation is maintained in neurons and astrocytes as both are known to be derived from NPCs. We have discussed this issue in the text. As suggested by the reviewers, we performed immunohistochemistry assays using neuron/astrocyte-specific antibodies to confirm that the differentiation of neurons and astrocytes in *Stag2*-depleted brains were largely normal (Figure 2—figure supplement 1B–E).

2. More details are required for the 3D chromatin structure analysis. Although the data are mainly in line with the literature, opposite to other work (Wutz et al., EMBOJ 2017; Pekowska et al., Nature 2017; Casa et al., Genome Res 2020), the authors did not find any TAD alterations. This might be related to the resolution used; the methods or legends do not state which resolution has been used. Please clarify and include this information in the figure legend, and make sure that at least 25 kb resolution is used for TAD analysis. You should report a summary of the sequencing data including the total reads obtained and, importantly the number of valid cis-pairs in each of their libraries etc.

For the TAD analysis, we used 40 kb resolution in our Hi-C analysis pipeline. This information has now been added to the figure legends. In our *Stag2*-depleted oligodendrocytes, we did not find large changes in TADs. As suggested by the reviewers, we re-analyzed our Hi-C datasets at 25 kb and 10 kb resolutions. As shown in Figure 6G and Figure 6—figure supplement 1D, consistent with our original analyses, there was minimal TAD alteration in *Stag2*-depleted oligodendrocytes. Our initial TAD analysis revealed that a few TADs had altered intra-TAD activities. We did not, however, find a correlation between changes in intra-TAD activities and the gene expression changes for genes located inside these TADs. A summary of Hi-C statistics is now included in Figure 6—figure supplement 1E.

3. Also for the Hi-C data: from Figure 5F and 5E, the correlation between the 2 biological replicates seems to be not very high; please comment. Since the effect is only on the loops and is quite modest, it is important that you demonstrate that it is reproducible and report whether this effect was seen in independent datasets, and whether the same loops were affected in each case.

In the original Figure 5E,F, the PCA and similarity analyses were performed using the insulation score and TAD boundary information, respectively. After the re-analysis of our Hi-C data with higher resolutions, we conducted similar analyses. The results were plotted in Figure 6—figure supplement 1. Overall, the analyses showed that there was high similarity between replicates, especially between the wild-type samples. As the insulation score and TAD boundary information were not ideal parameters to assess sample reproducibility, we calculated the reproducibility score of our replicate samples using HiCrep (v1.11.0) (Yang et al., 2017) at 25 kb resolution (Figure 6—figure supplement 1A). The stratum-adjusted correlation coefficient (SCC) is above 97% for wild-type samples and above 98% for *Stag2*-depleted samples for most chromosomes, except the Y chromosome. This score is comparable to previous publications (Hsieh et al., 2020; Li et al., 2020). Thus, we conclude that there is high reproducibility for our biological replicates. The relatively low SCC (above 75%) for the Y chromosome is likely due to fewer valid interactions within this chromosome in our dataset.

In the initial submission, we showed a reduction of loop numbers at gene promoters, including myelin gene promoters in *Stag2*-depleted oligodendrocytes. The reviewers wanted to know whether this reduction was reproducible in independent datasets. To address this question, we performed an aggregated peak analysis (APA) using GENOVA (van der Weide et al., 2021) on our replicates of Hi-C matrices in Figure 7—figure supplement 1B–D. The biological replicates showed good reproducibility on called loops, as well as genotype-specific loops. Examples of loop reduction at myelination genes for the duplicated Hi-C matrices are shown in Figure 7—figure supplement 2D.

4. Regarding the single-cell and purified OL RNAseq, only two biological replicates have been used, but three would be recommended.

It is common practice to use one or two transcriptomic libraries for each genotype of mice in single-cell transcriptomic analyses (Chang et al., 2022; Fang et al., 2019; Yang et al., 2020). We feel that two libraries for each genotype of mice are adequate, as the results from the two biological replicates are highly similar. For the bulk RNA-seq from purified OLs, we enriched primary oligodendrocytes using MACS system. Brain samples from 3-4 P12-14 pups were pooled together during the purification of primary cells. Although there were only two replicates of each genotype, each replicate sample contained biological materials from multiple mouse brains. Furthermore, the results of OL samples were consistent with the whole-brain RNA-seq data, which were obtained with more biological replicates.

5. Please provide a better discussion of what distinguishes genes that are or are not regulated by STAG2. On page 15, you speculate that STAG2 might interact with oligodendrocyte-specific transcription factors and be preferentially recruited to myelination genes. Preferential recruitment to myelination genes should be tested using the existing ChIP-seq data represented in Figure S8.

As suggested by reviewers, we integrated our RNA-seq analysis of OLs with the

STAG2 ChIP-seq results (Supplementary File 2). In our pathway analysis using both IPA (Figure 4—figure supplement 4E) and gene ontology (Figure 4—figure supplement 5 and 6), the axon ensheathment and oligodendrocyte differentiation-related genes were enriched in the down-regulated group, but not in the up-regulated group. Among the 271 >2-fold downregulated genes, there were 210 genes (77%) with STAG2 enrichment near the transcriptional start site (TSS ± 2 kb). Only 117 out of the 292 >2-fold upregulated genes (40%) had STAG2 ChIP-seq peaks at their TSS ±2 kb regions. Thus, STAG2 occupied many oligodendrocyte-specific gene promoters. Representative plots of STAG2 tracks at genomic loci of cholesterol biosynthesis genes and myelin genes were shown in Figure 5D and Figure 5—figure supplement 1, respectively.

6. Similarly, all three of the models shown in Figure 7 appear to indicate that STAG2 ordinarily regulates transcription through the formation of promoter-anchored loops, via a mechanism that involves direct binding to the relevant promoter in question. STAG2 binding sites should therefore be enriched at promoters where loops are lost in the STAG2 mutant condition. You should determine whether this is the case and discuss the implications for their models if it is not.

We hypothesized that STAG2-formed loops at relevant gene promoters facilitate their transcription. As the reviewers mentioned, it is important to test if STAG2 was enriched at gene promoters that were anchored with loops lost in the *Stag2*-depleted OLs. We integrated the STAG2 ChIP-seq data and the RNA-seq analysis with the total loops called from the OL samples (Supplementary File 2). As mentioned above, among the 271 >2-fold downregulated genes, 77% of genes were bound with STAG2 near their promoters (TSS ± 2 kb). Among the 162 downregulated genes with reduced promoter-anchored loops in the *Stag2*-depleted cells, 137 genes (85%) had STAG2 peaks at their promoters (TSS ± 2kb). These results are consistent with the proposed model.

7. Similarly, you hypothesise three possible ways for Stag2 mediated gene expression regulation. One is by assisting enhancer-promoter loops. The authors have generated H3K27ac Chip from purified OLs, and this could be used to map E-P loops and test their hypothesis.

We hypothesized that STAG2 might extrude promoter-anchored loops (P-loops) at myelination genes to enhance their transcription during development. As suggested by the reviewers, we examined whether the enhancer-promoter (E-P) loops were perturbed by the loss of STAG2. Using the H3K27ac peaks from OLs, we defined the enhancer regions by filtering out the peaks overlapping with promoters and identified E-P loops (Author response image 1). Specifically, promoter regions were defined as ±2kb from TSS for all transcripts in the UCSC mm10 mouse reference genome. The H3K27ac narrow peaks called by MACS2 (v2.0.10) with a q-value cut-off of 0.05 was used to define the enhancer region after removing the peaks overlapping with the promoter region. Enhancer-promoter loops were identified if one end was anchored to a H3K27ac peak, and the opposite end was anchored to a promoter region. E-P loops anchored genes from *Stag2^f/y^* and *Stag2^f/y^;Nes^Cre^* cells were compared and classified into “*Stag2^f/y^*only”, “*Stag2^f/y^;Nes^Cre^*-only”, and “Common” categories. The significant active genes (logCPM >0, FDR < 0.05) were used for gene expression comparison analysis. Gene expression changes of these categories were compared (Figure-for-reviewers 1C).

We found that genes only associated with E-P loops in the *Stag2-*depleted cells were more activated, compared to genes associated with E-P loops in both *Stag2^f/y^* and *Stag2^f/y^;Nes^Cre^* OLs. In other words, STAG1-mediated E-P loops might be more conducive for gene activation. During the revision process of our manuscript, a newly published study (Chu et al., 2022) showed that H3K27ac increased at STAG1-STAG2 switch sites and enhanced loop anchors in *STAG2*-knockdown melanoma cells. Consistent with this published study, our ChIP-seq analysis indicated a slight increase of H3K27ac binding at the promoters of up-regulated genes, but not at promoters of stable or down-regulated genes in *Stag2*-depleted cells (Figure 5B). STAG1 substitution in *Stag2*-depleted cells might induce ectopic E-P loops formation, elevated level of H3K27ac, and gene activation. Alternatively, gained E-P loops could result from weakened TAD boundary insulation as proposed previously (Lupianez et al., 2015).

We did not find that the genes losing E-P loops in the *Stag2*-depleted cells became more repressed than “Common” genes. Thus, loss of direct contact with non-promoter distal genomic regulatory elements do not always lead to transcription repression in *Stag2*-depleted cells. About 17% of the down-regulated genes had lost E-P loops in the *Stag2*-depleted OLs (Author response image 1), but we did not find overrepresentation of myelination or cholesterol biosynthesis pathways in “*Stag2^f/y^*-only” group by GO analysis. Thus, loss of E-P loops is not the only underlying reason for the repression of myelination genes in *Stag2*-depleted cells.

There are, however, several major caveats with our analysis. One caveat is that H3K27ac peaks alone are not sufficient to define active enhancers. ChIP-seq experiments of enhancer binding transcription factors would provide more accurate features for identifying cell-type specific enhancer elements or “Super-enhancers” (Whyte et al., 2013). Another caveat is that, when we filtered out H3K27ac peaks overlapping with promoters, the promoter-to-promoter (PP) loops were left out. Active gene promoters, when interacting with each other, could also potentially serve as active genomic regulatory elements (Bonev et al., 2017; Chepelev et al., 2012; Hsieh et al., 2020). The contribution of STAG2-mediated P-P loops to the transcriptional regulation of OL genes was thus neglected from our H3K27ac peaks-based analysis. Due to these major caveats, we are not confident about the validity of our analysis and do not wish to include this analysis in the revised manuscript. This issue will be further investigated in the lab in future studies. We hope the reviewers would understand.

**Author response image 1. sa2fig1:** Enhancer-promoter loops and E-P loop-anchored gene expression change. (A) Scheme for identifying enhancer-promoter (E-P) loops. H3K27ac peaks at gene TSS ±2kb region are removed. E-P loops are anchored to promoters on one end and to the remaining H3K27ac peaks on the other end. (B) Aggregate peak analysis of E-P loops identified from *Stag2^f/y^* or *Stag2^f/y^;Nes^Cre^* OLs on the 10 kb Hi-C matrices using GENOVA. (C) Boxplot of the expression levels for genes in the indicated categories. *Stag2^f/y^*, genes only anchored to E-P loops in *Stag2^f/y^* cells; *Stag2^f/y^;Nes^Cre^*, genes only anchored to E-P loops in *Stag2^f/y^;Nes^Cre^* cells; Common, genes anchored to E-P loops in both *Stag2^f/y^* and *Stag2^f/y^;Nes^Cre^* cells. Significant active genes with FDR<0.05 were included in the analysis. Unpaired Wilcoxon test was used for the statistical analysis. **p < 0.01, ****p < 0.0001, ns, not significant. (D) Venn diagram showing the comparison of down-regulated gene list, up-regulated gene list in *Stag2^f/y^;Nes^Cre^* OLs, and the EP loop-anchored active gene lists specific to *Stag2^f/y^* or *Stag2^f/y^;Nes^Cre^* OLs, respectively. Gene counts are shown. (E) Snapshots of the contact maps of genomic locus of repressed genes with lost E-P loops and H3K27ac peaks in the *Stag2^f/y^;Nes^Cre^* cells. Tracks and narrow peaks from STAG2 and H3K27ac ChIP-seq as well as the loops are plotted below. Genes of interest are highlighted in red. The transcription directions are indicated by the arrows. Lost loop anchors and H3K27ac peaks are framed with grey dashed lines. (F) A snapshot of the contact maps of genomic locus of an activated gene *Cybrd1* with gained E-P loop in the *Stag2^f/y^;Nes^Cre^* cells. Tracks and narrow peaks from STAG2 and H3K27ac ChIP-seq as well as the loops are plotted below. The *Cybrd1* gene body and TSS are highlighted in red. The gained loop anchor is framed with grey dashed lines.

8. Figure 6D – A gene with a better-defined role in myelination than Pls1 would be preferable to use as an exemplar locus here.

As the reviewers suggested, we have replaced the original graph with the Hi-C matrix plotting of a genomic region surrounding *Mal*, a gene encoding a proteolipid localized in compact myelin. It has a well-established function in myelin biogenesis.

9. Please provide the list of DE genes, STAG2 genomic occupancy, or promoters anchored loops as a supplementary file. This will help the readability of the paper and will allow full disclosure of the dataset to the community. Furthermore, please provide the reviewer passkey for the GEO link.

As the reviewers suggested, we have added Supplementary File 2 that included the list of differential-expressed gene ID, the distance of nearby STAG2 peaks to their TSS, and the promoter-anchored loops in the wildtype and *Stag2*-depleted samples. To review GEO accession GSE186894, go to https://www.ncbi.nlm.nih.gov/geo/query/acc.cgi?acc=GSE186894 and enter token almhoaomxhkntsz into the box.

10. The authors are recommended to perform GO and pathway analysis using some other tools in addition to IPA, like GSEA. Also, since the pathways shown in the chart in Fig2Da and 4G are the top enriched pathways, it would be useful if the authors could provide a list of all the enriched pathways. Furthermore, please clarify what you used as background for the enrichment analysis.

As the reviewers suggested, we performed over-representation analysis for gene ontology (GO) using ClusterProfiler (Wu et al., 2021) in addition to the IPA analysis. The results were plotted in Figure 4—figure supplement 5 and 6. Ensheathment of neurons and gliogenesis were among the top over-represented biological pathways from downregulated genes in both datasets, consistent with a function of STAG2 in regulating myelin-related genes. The complete list of the enriched pathways related to the original Figure 2D and 4G are now provided in Supplementary File 1 and Figure 2—figure supplement 2. The complete gene lists were used as background for the enrichment analysis.

11. From the purified OL RNAseq, the authors found 271 down and 292 upregulated genes. Please clarify what these genes are. In Fig4G, you showed a pathway analysis of all the DE, and it is not clear which gene set has been used for FigS7E. It would be useful to characterize the UP and DOWN genes separately.

The down- and up-regulated genes are now listed in Supplementary File 2. The down-regulated genes of OL RNA-seq experiment were used for Ingenuity Pathway analysis in Figure S7E. We also did GO analysis using ClusterProfiler for the UP and DOWN genes separately in Figure 4—figure supplement 5 and 6. The down-regulated genes were enriched for oligodendrocyte differentiation and cholesterol metabolic process, while the up-regulated genes were enriched for cilium assembly and microtubule formation.

12. In the single-cell RNAseq, you have used nFeature_RNA(200-9500). This seems to be a very high threshold, with the risk to include cell doublets. Please disclose whether you can reproduce the same analysis using a lower threshold.

Based on the previous literature, a median number of about 7,000 genes can be detected per neuron by the Chromium Single Cell kit (10x Genomics, v3) (Armand et al., 2021; Yao et al., 2021), which was used in our study. We feel that the nFeature_RNA (200-9500) threshold is reasonable. On the other hand, we agree with the reviewers that our criteria may risk including some cell doublets. We thus used a lower threshold [nFeature_RNA (200-6500)] to reanalyze the data (Author response image 2). Most cell clusters from the initial analysis were rediscovered and the initial fundings were confirmed. We thus decided to keep the original figure.

13. It would be useful to test using the Stag2 ChIP data in isolated OLs, whether loop loss is observed preferentially on Stag2 occupied sites and how STAg2 occupancy correlates with loop score of up/down/no DE genes.

To check if loop loss was enriched at gene promoters occupied by STAG2, we integrated our STAG2 ChIP-seq, RNA-seq results of OLs, and loops anchored at DEGs and stable gene promoters. STAG2 occupied 90% (3,333 out of 3,689) gene promoters, which contained anchors of *Stag2^f/y^*-specific loops (i.e. lost loops). Thus, in most cases, loop loss does occur at STAG2 occupied sites. In the original submission, we showed that STAG2 is preferentially enriched at promoters of the down-regulated and stable genes, whereas it is less enriched at the promoters of up-regulated genes (Figure 5C). In our integration analysis, the down-regulated genes and the stable genes have frequent STAG2 occupancy at promoters anchored with loops, but a lower percentage of the up-regulated genes is occupied by STAG2 near the TSS. It is also consistent with the loop score comparison, which shows up-regulated gene promoters are associated with loops of lower scores (Figure 7—figure supplement 3B). We further compared the loop scores of loops anchored at DEGs with or without STAG2 enrichment (Figure 7—figure supplement 3C,D). The loops anchored at down-regulated genes with STAG2 binding had significantly higher loop scores, compared to those with no STAG2 binding. Surprisingly, this difference was still observed in *Stag2*-deleted cells, suggesting that the stronger looping observed at these gene promoters might be independent of STAG2 binding. By contrast, the loops anchored at up-regulated genes with STAG2 binding had lower loop scores. These differences became insignificant in the *Stag2*-deleted cells. The loop scores of loops anchored at stable genes were not affected by STAG2 occupancy.

**Author response image 2. sa2fig2:** Transcriptome analysis of *Stag2*-depleted forebrains. (A) *t*-SNE plot of cell clusters with the filtering criteria of nFeature_RNA (200-6500) in the single-cell analysis by Seurat. (B) *t*-SNE clustering as in (A) but colored by genotype. (C) Left panel: cell type composition and percentage as colored in (A). Right panel: percentage of cell clusters of the oligodendrocyte lineage. (D) Heatmap showing the expression levels of cell-type signature genes. (E) FeaturePlot of the representative genes (*Mal* and *Nkx6-2*) specifically suppressed in the late stages of OLs in *Stag2*-depleted forebrains. A maximum cutoff of 3 was used.

14. A more detailed Methods section is required: for example, how many replicates for the sequencing, sequencing depth, the sequence of the probes, PCR primers, antibodies (codes and quantity); more details for the OL isolation procedure. Methods for Cas9 mediated KO are missing.

As the reviewers suggested, we have added more details to the Methods section. The information of sequencing details is included in Figure 6—figure supplement 1E. The primer sequences, probe sequences, and the information of antibodies and software used in this manuscript are now included in the Key Resources Table.

15. Similarly, Figure Legends need more details for clarity

As suggested by the reviewers, we have added more details to the figure legends.

References

Armand, E.J., Li, J., Xie, F., Luo, C., and Mukamel, E.A. (2021). Single-Cell Sequencing of Brain Cell Transcriptomes and Epigenomes. Neuron *109*, 11-26.

Bonev, B., Mendelson Cohen, N., Szabo, Q., Fritsch, L., Papadopoulos, G.L., Lubling, Y., Xu, X., Lv, X., Hugnot, J.P., Tanay, A.*, et al.* (2017). Multiscale 3D Genome Rewiring during Mouse Neural Development. Cell *171*, 557-572 e524.

Chang, C.S., Yu, W.H., Su, C.C., Ruan, J.W., Lin, C.M., Huang, C.T., Tsai, Y.T., Lin, I.J., Lai, C.Y., Chuang, T.H.*, et al.* (2022). Single-cell RNA sequencing uncovers the individual alteration of intestinal mucosal immunocytes in Dusp6 knockout mice. iScience *25*, 103738.

Chepelev, I., Wei, G., Wangsa, D., Tang, Q., and Zhao, K. (2012). Characterization of genomewide enhancer-promoter interactions reveals co-expression of interacting genes and modes of higher order chromatin organization. Cell Res *22*, 490-503.

Chu, Z., Gu, L., Hu, Y., Zhang, X., Li, M., Chen, J., Teng, D., Huang, M., Shen, C.H., Cai, L.*, et al.* (2022). STAG2 regulates interferon signaling in melanoma via enhancer loop reprogramming. Nat Commun *13*, 1859.

Fang, X., Huang, L.L., Xu, J., Ma, C.Q., Chen, Z.H., Zhang, Z., Liao, C.H., Zheng, S.X., Huang, P., Xu, W.M.*, et al.* (2019). Proteomics and single-cell RNA analysis of Akap4-knockout mice model confirm indispensable role of Akap4 in spermatogenesis. Dev Biol *454*, 118-127. Hsieh, T.S., Cattoglio, C., Slobodyanyuk, E., Hansen, A.S., Rando, O.J., Tjian, R., and Darzacq, X. (2020). Resolving the 3D Landscape of Transcription-Linked Mammalian Chromatin Folding. Mol Cell *78*, 539-553 e538.

Li, Y., Haarhuis, J.H.I., Sedeno Cacciatore, A., Oldenkamp, R., van Ruiten, M.S., Willems, L., Teunissen, H., Muir, K.W., de Wit, E., Rowland, B.D.*, et al.* (2020). The structural basis for cohesin-CTCF-anchored loops. Nature *578*, 472-476.

Lupianez, D.G., Kraft, K., Heinrich, V., Krawitz, P., Brancati, F., Klopocki, E., Horn, D., Kayserili, H., Opitz, J.M., Laxova, R.*, et al.* (2015). Disruptions of topological chromatin domains cause pathogenic rewiring of gene-enhancer interactions. Cell *161*, 1012-1025.

van der Weide, R.H., van den Brand, T., Haarhuis, J.H.I., Teunissen, H., Rowland, B.D., and de Wit, E. (2021). Hi-C analyses with GENOVA: a case study with cohesin variants. NAR Genom Bioinform *3*, lqab040.

Whyte, W.A., Orlando, D.A., Hnisz, D., Abraham, B.J., Lin, C.Y., Kagey, M.H., Rahl, P.B., Lee, T.I., and Young, R.A. (2013). Master transcription factors and mediator establish super-enhancers at key cell identity genes. Cell *153*, 307-319.

Wu, T., Hu, E., Xu, S., Chen, M., Guo, P., Dai, Z., Feng, T., Zhou, L., Tang, W., Zhan, L.*, et al.* (2021). clusterProfiler 4.0: A universal enrichment tool for interpreting omics data. Innovation (N Y) *2*, 100141.

Yang, C., Siebert, J.R., Burns, R., Zheng, Y., Mei, A., Bonacci, B., Wang, D., Urrutia, R.A., Riese, M.J., Rao, S.*, et al.* (2020). Single-cell transcriptome reveals the novel role of T-bet in suppressing the immature NK gene signature. *ELife 9*.

Yang, T., Zhang, F., Yardimci, G.G., Song, F., Hardison, R.C., Noble, W.S., Yue, F., and Li, Q. (2017). HiCrep: assessing the reproducibility of Hi-C data using a stratum-adjusted correlation coefficient. Genome Res *27*, 1939-1949.

Yao, Z., Liu, H., Xie, F., Fischer, S., Adkins, R.S., Aldridge, A.I., Ament, S.A., Bartlett, A., Behrens, M.M., Van den Berge, K.*, et al.* (2021). A transcriptomic and epigenomic cell atlas of the mouse primary motor cortex. Nature *598*, 103-110